

# Physical representations for scattering amplitudes and the wavefunction of the universe

**Paolo Benincasa**[⋆] **and William J. Torres Bobadilla**[†]

Max-Planck-Institut für Physik, Werner-Heisenberg-Institut, 80805 München, Germany

⋆ pablowellinhouse@anche.no, † torres@mpp.mpg.de

## Abstract

The way we organise perturbation theory is of fundamental importance both for computing the observables of relevance and for extracting fundamental physics out of them. If on one hand the different ways in which the perturbative observables can be written make manifest different features (*e.g.* symmetries as well as principles such as unitarity, causality and locality), on the other hand precisely demanding that some concrete features are manifest lead to different ways of organising perturbation theory. In the context of flat-space scattering amplitudes, a number of them are already known and exploited, while much less is known for cosmological observables. In the present work, we show how to systematically write down both the wavefunction of the universe and the flat-space scattering amplitudes, in such a way that they manifestly show physical poles only. We make use of the invariant definition of such observables in terms of *cosmological polytopes* and their *scattering facet*. In particular, we show that such representations correspond to triangulations of such objects through hyperplanes identified by the intersection of their facets outside of them. All possible triangulations of this type generate the different representations. This allows us to provide a general proof for the conjectured all-loop causal representation of scattering amplitudes. Importantly, all such representations can be viewed as making explicit a subset of compatible singularities, and our construction provides a way to extend Steinmann relations to higher codimension singularities for both the flat-space scattering amplitudes and the cosmological wavefunction.



# 1  Introduction

Our ability to understand physical phenomena is intimately tied to our capacity of describing them in terms of observables, understand the analytic structure of the latter and, finally, compute them. In particular, the analytic structure of relevant observables is highly constrained by the basic principles of unitarity and causality as well as, in the case of accessible high-energy processes in asymptotically flat space-times, by locality and Lorentz invariance. Furthermore, the way we represent them can make some of these principles manifest, or hide them in favour of other features.

In perturbation theory for flat-space particle scattering, Feynman diagrammatics constitutes the standard textbook approach. It makes unitarity and locality manifest at the expenses of introducing unphysical degrees of freedom and, consequently, gauge and field redefinition redundancies, leading to cumbersome expressions. The complexity of final results is, however, unnecessary and just a by-product of the diagrammatics itself. In effect, scattering amplitudes written in terms of on-shell data only [1–7], *i.e.* just in terms of physical degrees of freedom and in an intrinsically gauge-invariant way, their simplicity would become manifest. The price to pay is the lost of manifest locality, as spurious poles are introduced, and manifest unitarity, as just a subset of factorisation channels are manifest, while the missing one appears as a soft singularity [8].

Feynman and on-shell diagrammatics are just two ways to organise perturbation theory, but indeed they are not the only possible ways. Old-fashioned perturbation theory (OFPT) [9,10] clarifies the singularity structure in scattering amplitudes and their relation to intermediate states [9,11], at the price of losing manifest Lorentz invariance and time-translation invariance at intermediate stages, and it is related to Feynman diagrammatics via Feynman's tree theorem [10,12]; The loop-tree duality formulation (LTD) [13,14] expresses loop amplitudes in terms of tree-level-like objects, by simultaneously cutting, or setting on shell, one propagator per loop in a given Feynman graph, and appears to be a suitable approach to perform numerical evaluations of scattering amplitudes [15,16]. The extensive use of LTD, motivated by novel formulations [17–20], has led to a representation that only displays physical information, the so-called causal representation (CR) [21–24], which overcomes numerical instabilities that take place in expressions generated from LTD. In view of the simplicity of analytic expressions in CR, this representation has been conjectured to hold at all-loop orders [25,26].

Which representation is more suitable depends on the specific question we are asking: all ways of organising perturbation theory allow us to understand and exploit different aspects of the physics encoded in flat-space scattering amplitudes and their computation, since they yet represent the very same quantity. It would be, then, desirable to have an invariant way of understanding scattering amplitudes and relate these different representations.



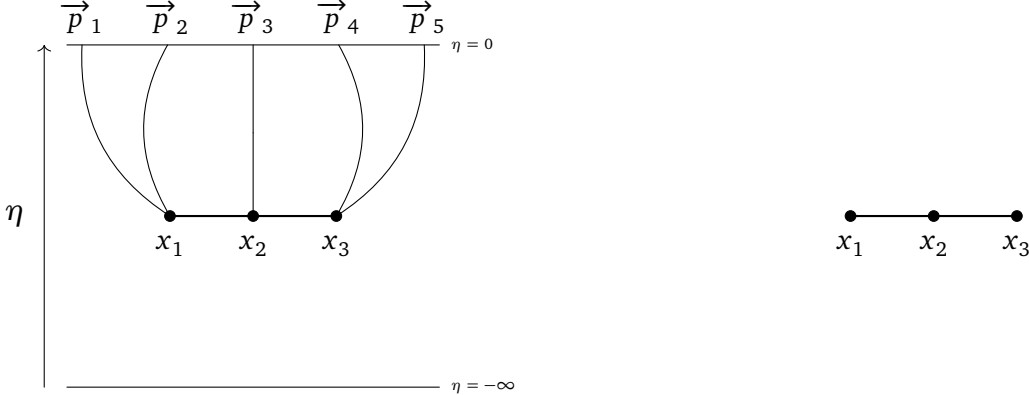

Figure 1: From Feynman to reduced graphs. A Feynman graphs (on the left) that contributes to the wavefunction of the universe, can be mapped into a reduced graph (on the right) by suppressing its external edges [34].

In cosmology, the understanding of relevant quantum mechanical observables, such as the spatial correlations and the wavefunction of the universe, as well as the physics they encode and the technology to compute them, are all significantly more primitive. The way that fundamental physical principles, such as unitarity and causality, reflect into and constrain these observables, started to be elucidated only recently, via the proof of a cosmological optical theorem [27] and the associated cutting rules [28–31], the positivity of the spectral density [32,33], as well as the proof of existence Steinmann-like relations [34].

Perturbation theory has been usually organised in terms of Feynman-like graphs computed in the in-in formalism for correlation functions, or directly in the Feynman representation for the wavefunction coefficients. Just recently, new representations have been used for the computation of individual Feynman graphs, such as: OFPT [35], which is expressed in terms of physical singularities only; a representation for exchange graphs which is obtained by imposing symmetries, singularities, and factorisation properties [36,37]; the Mellin-Barnes representation [38–41]; as well as the cosmological optical theorem and the cutting rules allow for new representations making unitarity manifest. As for the scattering amplitude case, it is customary to ask whether there exist an invariant way to understand the wavefunction of the universe, and how all representations are related to each other.

In this paper, we argue how these questions for both scattering amplitudes and the wavefunction of the universe not only have answers, but they can be given at once as they rely on the same mechanism.

An invariant description for the wavefunction of the universe as well as the flat-space scattering amplitudes, at least for a large class of toy models of scalars with polynomial interactions, is provided by the *cosmological polytopes* [35,42]. They are geometrical-combinatorial objects with an intrinsic mathematical definition that make no reference to physical concepts such as a Hilbert space and space-time, and yet they turn out to encode all the properties we ascribe to the wavefunction of the universe. A cosmological polytope is in a one-to-one correspondence with a *canonical form, i.e.* a differential form with logarithmic singularities on, and only on, the boundaries of the polytope itself[1] [35]. Its *canonical function*, obtained from the canonical form by stripping out the standard measure of the projective space where the polytope is defined, precisely returns the contribution of a Feynman graph $\mathcal{G}$ to the wavefunction. Consequently, singularities along the boundaries of the polytope are in one-to-one

---

[1]This is a generic property of any positive geometry [35] and the cosmological polytope is just very concrete example of them.

correspondence with singularities of the wavefunction itself.

One of them is the locus in kinematic space where the sum of the moduli of the spatial momenta $\{\vec{p}_i\}_{i=1}^n$ of all the states, $E_{\text{tot}} := \sum_{i=1}^n |\vec{p}_i|$, vanishes. As all $|\vec{p}_i|$ are positive for physical processes, such a locus can be reached only upon analytic continuation outside of the physical region, allowing some states to have $|\vec{p}_i|$ positive and some other negative: as it is approached, the wavefunction reduces to the high-energy limit of the flat-space scattering amplitudes [43–45] — if the scattering amplitude is trivial or simply the states in consideration do not have a flat-space counterpart, the coefficient of this singularity is a pure cosmological effect and the singularity itself is softer [46]. In the cosmological polytope description, $E_{\text{tot}} = 0$ identifies a codimension-1 boundary, the *scattering facet*, which is still a polytope whose canonical form encodes the flat-space scattering amplitude [35]. It allows for a combinatorial characterisation of flat-space unitarity and Lorentz invariance: the codimension-1 boundaries of the scattering facet turns out to factorise into two lower dimensional scattering facets and a simplex encoding the Lorentz invariant phase-space measure, providing the cutting rules; a contour integral representation of the canonical form of the scattering facet instead makes Lorentz invariance manifest [47].

Furthermore, multiple singularities in the wavefunction and scattering amplitudes correspond to higher codimension faces of the associated polytope $\mathcal{P}$, where $\mathcal{P}$ is respectively the full cosmological polytope $\mathcal{P}_{\mathcal{G}}$ and its scattering facet $\mathcal{S}_{\mathcal{G}}$. The analysis of the codimension-2 faces of $\mathcal{P}$, allowed for a combinatorial proof of the flat-space Steinmann relations as well as Steinmann-like relations for the wavefunction [34], *i.e.* the statement that double discontinuities across partially-overlapping channels vanish in the physical region[2]. As a codimension-2 face of $\mathcal{P}$ is given by the intersection of two of its facets onto the polytope itself, Steinmann relations combinatorially translate into the statement that each pair of facets corresponding to partially-overlapping channels intersect on a codimension-2 hyper-surface *outside* the polytope.

A natural operation on any polytope is a triangulation, or more generally the polytope subdivision, *i.e.* its division into a collection of polytopes such that their interiors are disjoint and their orientations are compatible. Consequently, its canonical form can be written as the sum of canonical forms of elements of such a collection. As the canonical function of a cosmological polytope *is* the contribution of a graph $\mathcal{G}$ to the wavefunction of the universe, writing it as a sum of canonical functions of a certain collection of polytopes that triangulates the cosmological polytope corresponds to provide a representation for it. Turning the table around, representations for a given graph contribution to wavefunction of the universe are provided by all possible triangulations of the cosmological polytope. A similar statement for flat-space scattering amplitudes holds on the scattering facet. Hence, the analysis and characterisation of different representations for the wavefunction of the universe and the scattering amplitudes boils down to the analysis and characterisation of triangulations of the cosmological polytopes.

Importantly, given a polytope $\mathcal{P}$, that as before can be either a cosmological polytope $\mathcal{P}_{\mathcal{G}}$ associated to a graph $\mathcal{G}$ or its scattering facet $\mathcal{S}_{\mathcal{G}}$, regular triangulations of $\mathcal{P}$ introduce spurious boundaries. This translates in decomposing the canonical form $\omega(\mathcal{Y}, \mathcal{P})$ into sums of canonical forms with singularities along such boundaries, which have to cancel upon summation.

In this paper, we are mainly concerned with triangulations of $\mathcal{P}$ which do not introduce any spurious singularity in the canonical form and, consequently, generate all representations for our observables with physical poles only. Such a discussion for $\mathcal{P} = \mathcal{S}_{\mathcal{G}}$ will allow us to prove the all-loop causal representation conjectured in [25]. Generally speaking, these triangulations can be viewed as regular triangulations of the dual polytope $\tilde{\mathcal{P}}$, since vertices of the latter are precisely related to singularities of the wavefunction. We will show how

---

[2]While the flat-space Steinmann-relations are a consequence of causality [48–55], this is still not clear in Steinmann-like relations for the wavefunction.

they correspond to triangulations of the polytope $\mathcal{P}$ through the locus of intersections of its facets outside the polytope itself. In effect, by means of latter identification and classification, we will be able to elucidate in detail vanishing multiple discontinuities as well as provide a procedure for triangulating $\mathcal{P}$ through such intersections. Thus, finding representations for our observables with physical poles only.

In what follows, we first provide a brief review of the wavefunction of the universe and the aspects of the cosmological polytopes which will be relevant for our discussion. We then discuss the intersections of the facets of the cosmological polytope outside of the polytope itself, providing a way to characterise the locus they form and which determines the locus of the zeroes of the canonical form. Importantly, this corresponds to the analysis of the multiple discontinuities of the wavefunction, leading to the extension of the Steinmann-like relations to higher codimension singularities. This allows us to write down a large class of signed-triangulations with no spurious boundaries, that correspond to a large class of representations with physical poles only, most of which turn out to be novel. We further apply this analysis to the scattering facet, obtaining an even larger class of representations among which we can identify the causal representation conjectured in [25] providing a proof for it. While all these representations look very different from each other, both for the wavefunction coefficients and the scattering amplitudes, our combinatorial language allows us to treat them on the same footing and emphasising their common features: all of them make manifest one higher codimension zero and a subset of the Steinmann relations extended to higher codimension singularities.

Finally, in the appendices we provide explicit examples and derivations of physical representations for the wavefunction of the universe and scattering amplitudes.

## 2 Cosmological Polytopes in a Nutshell

Let us consider the action for a scalar with time-dependent polynomial interactions in a $(d+1)$-dimensional Minkowski space-time:

$$S[\phi] = -\int d^d x \int_{-\infty}^{0} d\eta \left[ \frac{1}{2}(\partial\phi)^2 - \sum_{k\geq 3} \frac{\lambda_k(\eta)}{k!} \phi^k \right]. \tag{1}$$

This class of scalar toy models contains a conformally-coupled scalar in Friedmann-Robertson-Walker (FRW) cosmologies, provided that the time-dependent coupling $\lambda_k(\eta)$ is identified to be

$$\lambda_k(\eta) = \lambda_k \vartheta(-\eta)[a(\eta)]^{(2-k)(d-1)+2}, \tag{2}$$

where $\lambda_k$ is a constant, while $a(\eta)$ is the time-dependent warp-factor for the Poincaré patch of the FRW cosmology metric $ds^2 = a^2(\eta)[-d\eta^2 + \delta_{ij}dx^i dx^j]$.

Quantum mechanical processes for this class of toy models can be described by the wave-function of the universe

$$\Psi[\Phi] = \mathcal{N} \int_{\phi(-\infty(1-i\varepsilon))=0}^{\phi(0)=\Phi} \mathcal{D}\phi \, e^{iS[\phi]}, \tag{3}$$

whose squared-modulus provides the probability distribution for the field configuration $\Phi$ at late-time. The boundary condition at early times selects the vacuum state, while the $i\varepsilon$-prescription regularises the path integral in such a region as it contains oscillatory phases and picks the positive frequency solution. Splitting $\phi$ into its free, classical mode $\phi_{\circ} = \Phi(\vec{p})e^{i|\vec{p}|\eta}$,

and its quantum fluctuation $\varphi$, in such a way that $\phi_\circ$ encodes the correct Bunch-Davies oscillatory behaviour at early times, $\varphi$ consequently has to satisfy vanishing boundary conditions at both early and late times. The bulk-to-boundary propagation is simply given by a positive frequency exponential.

A convenient way to treat the time-dependent coupling $\lambda_k(\eta)$ is to consider the following integral representation

$$\lambda_k(\eta) = \int_{-\infty}^{\infty} d\epsilon \, e^{i\epsilon\eta} \, \tilde{\lambda}_k(\epsilon), \tag{4}$$

with the exponential in (4) having the same form of a bulk-to-boundary propagator. Computing the perturbative wavefunction via Feynman integrals, it can be represented as the integral over an $\epsilon$ for each graph site[3] with measure $\tilde{\lambda}_k(\epsilon)$ of a universal integrand [35]: while the cosmology is completely fixed by the explicit form of the function $\tilde{\lambda}_k(\epsilon)$, such an integrand encodes features which are common to all models, and it will be the focus of our discussion.

Given a graph $\mathcal{G}$ with edges $\mathcal{E}$ and sites $\mathcal{V}$, the universal integrand $\psi_{\mathcal{G}}(x_s, y_e)$ is given by,

$$\psi_{\mathcal{G}}(x_s, y_e) = \int_{-\infty}^{0} \prod_{s\in\mathcal{V}} \left[ d\eta_s \, e^{ix_s\eta_s} \right] \prod_{e\in\mathcal{E}} G(y_e; \eta_{s_e}, \eta_{s'_e}), \tag{5}$$

where $x_s = \sum_{j\in s} |\vec{p}_j|$ is the sum of external energies[4] at a site $s \in \mathcal{V}$, $y_e$ is the energy of the state attached to the edge $e \in \mathcal{E}$, and $G$ is the bulk-to-bulk propagator satisfying the boundary condition that the fluctuations vanish at late time $\eta = 0$

$$G(y_e, \eta_{s_e}, \eta_{s'_e}) = \frac{1}{2y_e} \left[ e^{-iy_e(\eta_{s_e}-\eta_{s'_e})} \vartheta(\eta_{s_e} - \eta_{s'_e}) + e^{+iy_e(\eta_{s_e}-\eta_{s'_e})} \vartheta(\eta_{s'_e} - \eta_{s_e}) - e^{+iy_e(\eta_{s_e}+\eta_{s'_e})} \right]. \tag{6}$$

As the integrals defining $\psi_{\mathcal{G}}(x_s, y_e)$ only depend on the total energy $x_s$ at a site $s \in \mathcal{V}$, the universal integrand $\psi_{\mathcal{G}}(x_s, y_e)$ can be represented via a reduced graph, that is obtained by the original one suppressing the external lines (see Figure 1). Given a reduced graph we can assign the weight $x_s$ to each site $s \in \mathcal{V}$ and the weight $y_e$ to each edge $e \in \mathcal{E}$.

Any reduced graph $\mathcal{G}$ turns out to be in one-to-one correspondence with polytopes whose canonical form encodes precisely the universal wavefunction integrand $\psi_{\mathcal{G}}(x_s, y_e)$ [35]. First, any reduced graph can be seen as a set of two-site line graphs with some sites identifies. A given two-line graph, with weights $(x_i, y_i, x'_i)$ is associated to a triangle living in a projective space $\mathbb{P}^2$ with homogeneous local coordinates $\mathcal{Y} := (x_i, y_i, x'_i)$ and with the canonical basis of $\mathbb{R}^3$ in such coordinates, $\mathbf{x}_i = (1, 0, 0)$, $\mathbf{y}_i = (0, 1, 0)$, $\mathbf{x}'_i = (0, 0, 1)$, being the midpoints of its sides:

The vertices of such triangles are then identified by the triple of vectors

$$\{\mathbf{x}_i - \mathbf{y}_i + \mathbf{x}'_i, \, \mathbf{x}_i + \mathbf{y}_i - \mathbf{x}'_i, \, -\mathbf{x}_i + \mathbf{y}_i + \mathbf{x}'_i\}.$$

A collection of $n_e$ two-site line graphs thus corresponds to a collection of $n_e$ triangles living in $\mathbb{P}^{3n_e-1}$. The identification of the sites of the two-line graphs that maps this collection into a single connected graph $\mathcal{G}$ corresponds then to identifying the related triangles in the midpoints of their sides: the convex hull of the vertices of the triangles defines a polytope living in a

---

[3]In order to avoid language clash, we will use *site* to indicate the vertex of a graph, while *vertex* will be reserved for the highest codimension boundary of a polytope.

[4]With a bit abuse of language, we refer to the modulus of a spatial momentum as *energy*.

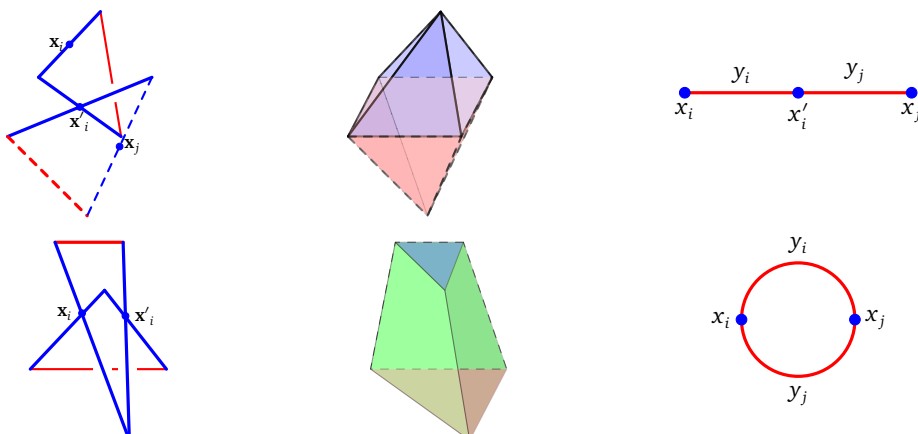

Figure 2: Visualisation of two cosmological polytopes. They are, respectively, obtained from the intersection of two triangles (on the left column) in one and two midpoints. The central and the right columns depict the corresponding convex hulls and the associated reduced graphs, respectively [34].

lower dimensional projective space with homogeneous local coordinates given by the weights $(x_s, y_e)$ associated to the sites and edges of $\mathcal{G}$. If $n_s$ and $n_e$ are respectively the number of sites and edges of $\mathcal{G}$, then the *cosmological polytope* defined in this way has $3n_e$ vertices and lives in $\mathbb{P}^{n_s+n_e-1}$ (see Figure 2).

Given a cosmological polytope $\mathcal{P}_\mathcal{G}$ associated to a reduced graph $\mathcal{G}$, its features are encoded into the associated canonical form

$$\omega(\mathcal{Y}, \mathcal{P}_\mathcal{G}) = \Omega(\mathcal{Y}, \mathcal{P}_\mathcal{G})\langle \mathcal{Y} d^N \mathcal{Y}\rangle, \tag{7}$$

which has logarithmic singularities on, and only on, all of its boundaries [35]. Its canonical function $\Omega(\mathcal{Y}, \mathcal{P}_\mathcal{G})$ turns out to be precisely the universal integrand $\psi_\mathcal{G}(x_s, y_e)$ and, consequently, the boundary structure of the cosmological polytope $\mathcal{P}_\mathcal{G}$ characterises the residues of $\psi_\mathcal{G}(x_s, y_e)$. Importantly, the *facets* of $\mathcal{P}_\mathcal{G}$ are in one-to-one correspondence with the connected subgraphs of $\mathcal{G}$: if $\mathfrak{g} \subseteq \mathcal{G}$ is a connected subgraph of $\mathcal{G}$, then the related facet of $\mathcal{P}_\mathcal{G}$ is given by the intersection $\mathcal{P}_\mathcal{G} \cap \mathcal{W}^{(\mathfrak{g})}$ between the cosmological polytope $\mathcal{P}_\mathcal{G}$ and a hyperplane identified by the dual vector

$$\mathcal{W}^{(\mathfrak{g})} = \sum_{s \in \mathcal{V}_\mathfrak{g}} \tilde{\mathbf{x}}_s + \sum_{e \in \mathcal{E}_\mathfrak{g}^{\text{ext}}} \tilde{\mathbf{y}}_e, \tag{8}$$

such that $\mathcal{Z}_i \cdot \mathcal{W}^{(\mathfrak{g})} = 0$ for all the vertices $\mathcal{Z}_i$ of $\mathcal{P}_\mathcal{G}$ belonging to $\mathcal{P}_\mathcal{G} \cap \mathcal{W}^{(\mathfrak{g})}$, while $\mathcal{Z}_i \cdot \mathcal{W}^{(\mathfrak{g})} > 0$ for all the other vertices of the cosmological polytope. Precisely these conditions fix the correspondence between facets and subgraphs as well as the form (8), where $\tilde{\mathbf{x}}_s$ and $\tilde{\mathbf{y}}_e$ are such that $\mathbf{x}_s \cdot \tilde{\mathbf{x}}_{s'} = \delta_{s's}$, $\mathbf{y}_e \cdot \tilde{\mathbf{y}}_{e'} = \delta_{e'e}$, $\mathbf{x}_s \cdot \tilde{\mathbf{y}}_e = 0$, and $\mathbf{y}_e \cdot \tilde{\mathbf{x}}_s = 0$. $\mathcal{V}_\mathfrak{g} \subseteq \mathcal{V}$ is the set of sites of $\mathfrak{g}$, and $\mathcal{E}_\mathfrak{g}^{\text{ext}}$ is the set of edges departing from $\mathfrak{g}$. Then

$$E_\mathfrak{g} := \mathcal{Y} \cdot \mathcal{W}^{(\mathfrak{g})} = \sum_{v \in \mathcal{V}_\mathfrak{g}} x_s + \sum_{e \in \mathcal{E}_\mathfrak{g}^{\text{ext}}} y_e, \tag{9}$$

is the total energy of the subprocess identified by the subgraph $\mathfrak{g}$, and we have $E_\mathfrak{g} \longrightarrow 0$ as the facet $\mathcal{P}_\mathcal{G} \cap \mathcal{W}^{(\mathfrak{g})}$ is approached.

In order to be able to identify the vertices of a cosmological polytope $\mathcal{P}_\mathcal{G}$ that are on the facet $\mathcal{P}_\mathcal{G} \cap \mathcal{W}^{(\mathfrak{g})}$ associated to a given subgraph $\mathfrak{g}$, it is convenient to introduce a marking ✖

which identifies those vertices $\mathcal{Z}_i$ of $\mathcal{P}_{\mathcal{G}}$ such that $\mathcal{Z}_i \cdot \mathcal{W}^{(\mathfrak{g})} > 0$, *i.e.* that *are not* on the facet

$$
\underset{\substack{x_i \quad y_e \quad x_i' \\ \mathcal{W} \cdot (\mathbf{x}_i - \mathbf{y}_e + \mathbf{x}_i') > 0}}{} \qquad
\underset{\substack{x_i \quad y_e \quad x_i' \\ \mathcal{W} \cdot (\mathbf{x}_i + \mathbf{y}_e - \mathbf{x}_i') > 0}}{} \qquad
\underset{\substack{x_i \quad y_e \quad x_i' \\ \mathcal{W} \cdot (-\mathbf{x}_i + \mathbf{y}_e + \mathbf{x}_i') > 0}}{} \quad .
$$

Given a subgraph $\mathfrak{g} \subseteq \mathcal{G}$, then the associated facet is identified by marking all the internal edges of $\mathfrak{g}$ in the middle, while all the edges departing from it close to the sites in $\mathfrak{g}$ [35].

Finally, the dual cosmological polytope $\tilde{\mathcal{P}}_{\mathcal{G}}$ of $\mathcal{P}_{\mathcal{G}}$ is defined as the convex hull identified by the vectors $\mathcal{W}^{(\mathfrak{g})}$ in the dual space of $\mathbb{P}^{n_s + n_e - 1}$, which is still $\mathbb{P}^{n_s + n_e - 1}$. Its facets are then associated to the co-vectors $\mathcal{Z}_i$ of the vertices of $\mathcal{P}_{\mathcal{G}}$. Therefore, the canonical function $\Omega(\mathcal{Y}, \mathcal{P}_{\mathcal{G}})$ of a cosmological polytope $\mathcal{P}_{\mathcal{G}}$ can be interpreted as the volume of $\tilde{\mathcal{P}}_{\mathcal{G}}$. Hence, the volume of $\tilde{\mathcal{P}}_{\mathcal{G}}$ provides the universal wavefunction integrand $\psi_{\mathcal{G}}(x_s, y_e)$.

## 3 Representations for the wavefunction

Let us consider a general graph $\mathcal{G}$ with $n_s$ sites and $n_e$ edges. The relations,

$$
\psi_{\mathcal{G}}(x_s, y_e) = \Omega(\mathcal{Y}, \mathcal{P}_{\mathcal{G}}) = \text{Vol}\{\tilde{\mathcal{P}}_{\mathcal{G}}\}, \tag{10}
$$

between the wavefunction contribution associated to $\mathcal{G}$ and the canonical function of the cosmological polytope $\mathcal{P}_{\mathcal{G}}$ on one side, and the volume of the dual cosmological polytope $\tilde{\mathcal{P}}_{\mathcal{G}}$ provide two complementary but related geometrical-combinatorial characterisation for $\psi_{\mathcal{G}}(x_s, y_e)$.

First, as any polytope, $\mathcal{P}_{\mathcal{G}}$ can be seen as the union of a collection $\{\mathcal{P}_{\mathcal{G}}^{(j)}\}$ of other polytopes $\mathcal{P}_{\mathcal{G}}^{(j)}$ such that any element $\mathcal{P}_{\mathcal{G}}^{(j)}$ of this collection is contained in $\mathcal{P}_{\mathcal{G}}$ and their interiors are disjoint. The collection $\{\mathcal{P}_{\mathcal{G}}^{(j)}\}$ provides a *triangulation* of the cosmological polytope $\mathcal{P}_{\mathcal{G}}$. As a consequence, the canonical function $\Omega(\mathcal{Y}, \mathcal{P}_{\mathcal{G}})$ can be written as the sum of the canonical functions of the elements of $\{\mathcal{P}_{\mathcal{G}}^{(j)}\}$ [56],

$$
\Omega(\mathcal{Y}, \mathcal{P}_{\mathcal{G}}) = \sum_{j=1}^{n} \Omega(\mathcal{Y}, \mathcal{P}_{\mathcal{G}}^{(j)}), \tag{11}
$$

in such a way that the singularities related to the common facets of the elements of the collection $\{\mathcal{P}^{(j)}\}$ cancel.

In the case of a cosmological polytope $\mathcal{P}_{\mathcal{G}}$, because of the equivalence (10) between its canonical function and the wavefunction, any triangulation of $\mathcal{P}_{\mathcal{G}}$ provides a representation for the wavefunction characterised by the presence of spurious poles, that precisely correspond to those boundaries of $\mathcal{P}_{\mathcal{G}}^{(j)}$ which *are not* boundaries of the cosmological polytope $\mathcal{P}_{\mathcal{G}}$ itself. Let us provide an interesting example of such a class of triangulations. Let us consider a cosmological polytope $\mathcal{P}_{\mathcal{G}}$ associated to a graph $\mathcal{G}$ with an external tree structure:

$$
\underset{\substack{x_1 \qquad x_2}}{\overset{y_{12}}{\bullet\!\!-\!\!-\!\!-\!\!\bullet}} \; \mathcal{G}' \quad .
$$

Focusing on such a tree structure, constituted by a 2-site subgraphs with weights $(x_1, x_2)$ for the sites and $y_{12}$ for the edge connecting them, it can be thought to be connected with $m_e \in [1, n_e - 1]$ other edges of the graph in the site $x_2$. This means that the convex hull of pairs of vertices $\{\mathbf{x}_j - \mathbf{y}_{j2} + \mathbf{x}_2, -\mathbf{x}_j + \mathbf{y}_{j2} + \mathbf{x}_2\}_{j=1}^{m_e + 1}$ define a polytope in $\mathbb{P}^{m_e}$ such that the $m_e + 1$ segments defined by such pairs intersect at their midpoint $x_2$. We can then triangulate the full $\mathcal{P}_{\mathcal{G}}$ by first triangulating such lower-dimensional polytope in such a way that the two vertices

$\{x_1 - y_{12} + x_2, -x_1 + y_{12} + x_2\}$ attached to the outer edge of the graph belong to different simplices:

$$\text{(12)}$$

where the marking $\circ$ identifies the vertices of the polytope associated to that edge. Then, the canonical function for the polytope $\mathcal{P}_{\mathcal{G}}$ can be written as the sum of the canonical functions related to the two polytopes $\mathcal{P}_{\mathcal{G}}^{(1)}$ and $\mathcal{P}_{\mathcal{G}}^{(2)}$ identified in the r.h.s. above:

$$\Omega(\mathcal{Y}, \mathcal{P}_{\mathcal{G}}) = \Omega(\mathcal{Y}, \mathcal{P}_{\mathcal{G}}^{(1)}) + \Omega(\mathcal{Y}, \mathcal{P}_{\mathcal{G}}^{(2)}). \tag{13}$$

From a purely graph perspective, this is equivalent to

$$\text{(14)}$$

which, in the case of a purely tree-level graph $\mathcal{G}'$ — i.e. the original graph $\mathcal{G}$ is taken to be tree level — then it provides the recursion relation proven in [35] via the frequency integral representation of the propagators.

The Feynman representation, which counts with $3^{n_e}$ terms for a graph $\mathcal{G}$ with $n_e$ edges, also can be obtained as a triangulation of the associated cosmological polytope $\mathcal{P}_{\mathcal{G}}$. However, it is not a regular triangulation as it involves vertices which *are not* vertices of $\mathcal{P}_{\mathcal{G}}$.

Notice that the facets of $\mathcal{P}_{\mathcal{G}}$ identifying the poles of $\psi_{\mathcal{G}}(x_s, y_e)$ are vertices of the dual cosmological polytope $\tilde{\mathcal{P}}_{\mathcal{G}}$. Hence, any regular triangulation of $\tilde{\mathcal{P}}_{\mathcal{G}}$ returns a representation for the wavefunction that involves only physical poles as it just uses the vertices of $\tilde{\mathcal{P}}_{\mathcal{G}}$. Therefore, at least in principle, classifying all the possible triangulations for a given dual cosmological polytope $\tilde{\mathcal{P}}_{\mathcal{G}}$ provides all the possible representation for the contribution to the wavefunction related to the related graph $\mathcal{G}$.

It is both interesting and useful to take the perspective of the actual cosmological polytope $\mathcal{P}_{\mathcal{G}}$. The locus $\mathcal{C}$ of the intersections of the facets of $\mathcal{P}_{\mathcal{G}}$ *outside* of $\mathcal{P}_{\mathcal{G}}$ identifies the zeroes of the canonical form $\omega(\mathcal{Y}, \mathcal{P}_{\mathcal{G}})$ and, hence, its numerator [57]. Then, the signed triangulations that do not generate spurious singularities in the canonical form are obtained considering a collection of polytopes $\{\mathcal{P}_{\mathcal{G}}^{(j)}\}$ such that, together with satisfying the usual conditions for triangulating $\mathcal{P}_{\mathcal{G}}$, the facets of each $\mathcal{P}_{\mathcal{G}}^{(j)}$ which are not facets of $\mathcal{P}_{\mathcal{G}}$ lie on such a locus. One example of such triangulations is given by OFPT, but the set of such signed triangulations is wider.

**Codimension-$2$ Intersections and Sequential Cuts**

The locus $\mathcal{C}$ of the intersections of the facets of $\mathcal{P}_{\mathcal{G}}$ outside $\mathcal{P}_{\mathcal{G}}$ is intimately related to those sequential cuts of the wavefunction which vanish. In particular, given a cosmological polytope $\mathcal{P}_{\mathcal{G}}$ associated to a graph $\mathcal{G}$, and given two hyperplanes $\mathcal{W}^{(\mathfrak{g}_1)}$ and $\mathcal{W}^{(\mathfrak{g}_2)}$ associated to the partially-overlapping subgraphs $\mathfrak{g}_1, \mathfrak{g}_2 \subset \mathcal{G}$, i.e. such that

$$\begin{aligned} \mathfrak{g}_1 \cap \mathfrak{g}_2 \neq \varnothing, \quad \mathfrak{g}_1 \cap \bar{\mathfrak{g}}_2 \neq \varnothing, \\ \bar{\mathfrak{g}}_1 \cap \mathfrak{g}_2 \neq \varnothing, \quad \bar{\mathfrak{g}}_1 \cap \bar{\mathfrak{g}}_2 \neq \varnothing, \end{aligned} \tag{15}$$

with $\bar{\mathfrak{g}}_j$ being the complement of $\mathfrak{g}_j$, then $\mathcal{P}_{\mathcal{G}} \cap \mathcal{W}^{(\mathfrak{g}_1)} \cap \mathcal{W}^{(\mathfrak{g}_2)} = \varnothing$ [34]. This implies that the double residue of the canonical form $\omega(\mathcal{Y}, \mathcal{P}_{\mathcal{G}})$ along the hyperplanes $\mathcal{W}^{(\mathfrak{g}_1)}$ and $\mathcal{W}^{(\mathfrak{g}_2)}$ vanishes [34]:

$$\text{Res}_{\mathcal{W}_{\mathfrak{g}_1}} \text{Res}_{\mathcal{W}_{\mathfrak{g}_2}} \omega(\mathcal{Y}, \mathcal{P}_{\mathcal{G}}) = 0. \tag{16}$$

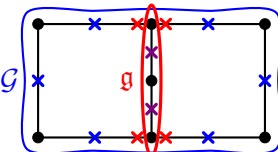 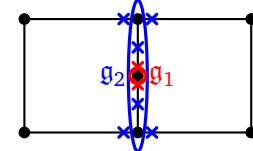

Figure 3: Examples of two facets related to subgraphs which do not correspond to partially overlapping channels but still their intersection lies outside of the cosmological polytope. They correspond to double cuts other than the ones which provide the Steinmann relations.

Because of the identification between the canonical function $\Omega(\mathcal{Y}, \mathcal{P}_\mathcal{G})$ and the wavefunction $\psi_\mathcal{G}$, the statement (16) implies Steinmann-like relations for $\psi_\mathcal{G}$:

$$\text{Res}_{E_{\mathfrak{g}_1}} \text{Res}_{E_{\mathfrak{g}_2}} \psi_\mathcal{G} = 0, \tag{17}$$

where $E_{\mathfrak{g}_j}$ is defined as in (9). The relation (17) can be promoted to a restriction on the double discontinuity of the tree-level $\Psi_\mathcal{G}$ as well as of the loop integrand of $\Psi_\mathcal{G}$ [34], as the integration of the site energies returns polylogarithms [35, 58].

Each hyperplane $\mathcal{W}^{(\mathfrak{g}_j)} \cap \mathcal{W}^{(\mathfrak{g}_k)}$ such that $\mathcal{P}_\mathcal{G} \cap \mathcal{W}^{(\mathfrak{g}_j)} \cap \mathcal{W}^{(\mathfrak{g}_k)} = \varnothing$ in codimension-2 defines a subspace of the locus $\mathcal{C}(\mathcal{P}_\mathcal{G})$ of the intersections of the facets of $\mathcal{P}_\mathcal{G}$ outside $\mathcal{P}_\mathcal{G}$. We can now ask two questions: how can we systematically identify all such intersections? And can we systematically define the different sets of basis which identify the locus $\mathcal{C}(\mathcal{P}_\mathcal{G})$?

A first observation is that there are other pairs of facets which intersect each other outside of $\mathcal{P}_\mathcal{G}$. Recall that facets identified by partially overlapping graphs (15) intersect each other on $\mathcal{P}_\mathcal{G}$ in a subspace of dimension

$$\dim(\mathcal{P}_\mathcal{G} \cap \mathcal{W}^{(\mathfrak{g}_1)} \cap \mathcal{W}^{(\mathfrak{g}_2)}) = n_s + n_e - 1 - \sum_{\mathcal{S}_\mathfrak{g}} 1, \tag{18}$$

where the sum runs over the scattering facets related to the subgraphs $\mathfrak{g}_1 \cap \mathfrak{g}_2$, $\mathfrak{g}_1 \cap \bar{\mathfrak{g}}_2$, $\bar{\mathfrak{g}}_1 \cap \mathfrak{g}_2$ [34]. For partially overlapping channels, all these intersections are non-empty, consequently $\mathcal{P}_\mathcal{G} \cap \mathcal{W}^{(\mathfrak{g}_1)} \cap \mathcal{W}^{(\mathfrak{g}_2)}$ has dimension $n_s + n_e - 4$ and the corresponding sequential cut vanishes.

Importantly, given two arbitrary subgraphs $\mathfrak{g}_1$, $\mathfrak{g}_2 \subseteq \mathcal{G}$, the graph $\mathcal{G}$ can be always thought to get divided into the intersections $\mathfrak{g}_1 \cap \mathfrak{g}_2$, $\mathfrak{g}_1 \cap \bar{\mathfrak{g}}_2$, $\bar{\mathfrak{g}}_1 \cap \mathfrak{g}_2$ and $\bar{\mathfrak{g}}_1 \cap \bar{\mathfrak{g}}_2$, and the intersection $\mathcal{P}_\mathcal{G} \cap \mathcal{W}^{(\mathfrak{g}_1)} \cap \mathcal{W}^{(\mathfrak{g}_2)}$ has the correct dimension if one of the first three is empty: this is the case for the two subgraphs are disjoint ($\mathfrak{g}_1 \cap \mathfrak{g}_2 = \varnothing$, $\mathfrak{g}_1 \cap \bar{\mathfrak{g}}_2 \neq \varnothing$ and $\bar{\mathfrak{g}}_1 \cap \mathfrak{g}_2 \neq \varnothing$), or one graph is contained in the other one (e.g. $\mathfrak{g}_2 \subset \mathfrak{g}_1$: $\mathfrak{g}_1 \cap \mathfrak{g}_2 = \mathfrak{g}_2 \neq \varnothing$, $\mathfrak{g}_1 \cap \bar{\mathfrak{g}}_2 \neq \varnothing$ and $\bar{\mathfrak{g}}_1 \cap \mathfrak{g}_2 = \varnothing$). Among these three configurations for the pair of subgraphs $\mathfrak{g}_1$ and $\mathfrak{g}_2$ (partially overlapping, disjoint and $\mathfrak{g}_i \subset \mathfrak{g}_j$), just the partially overlapping configuration satisfies the condition $\sum_{\mathcal{S}_\mathfrak{g}} 1 > 2$. However, there is one exception.

Let us consider $\mathfrak{g}_2 \subset \mathfrak{g}_1 \subseteq \mathcal{G}$ and let $n_{\mathfrak{g}_2}$ and $L_{\mathfrak{g}_1}$ be the number of edges departing from $\mathfrak{g}_2$ and the number of loops of $\mathfrak{g}_1$. Notice that $\mathfrak{g}_1$ identifies the scattering facet $\mathcal{S}_{\mathfrak{g}_1} := \mathcal{P}_\mathcal{G} \cap \mathcal{W}^{(\mathfrak{g}_1)}$ while $\mathfrak{g}_2$ identifies the facet $\mathcal{S}_{\mathfrak{g}_1} \cap \mathcal{W}^{(\mathfrak{g}_2)}$ of $\mathcal{S}_{\mathfrak{g}_2}$, were such an intersection be non-empty. However, if $n_{\mathfrak{g}_2} > L_{\mathfrak{g}_1}$ the number of vertices of $\mathcal{S}_{\mathfrak{g}_1} \cap \mathcal{W}^{(\mathfrak{g}_2)}$ is not enough to span the correct subspace [34] and the intersection $\mathfrak{g}_1 \cap \bar{\mathfrak{g}}_2 \neq \varnothing$ factorises in two lower-dimensional scattering facets, satisfying the condition $\sum_{\mathcal{S}_\mathfrak{g}} 1 > 2$ (see Figure 3). Hence

$$\text{Res}_{\mathcal{W}^{(\mathfrak{g}_1)}} \text{Res}_{\mathcal{W}^{(\mathfrak{g}_2)}} \omega(\mathcal{Y}, \mathcal{P}_\mathcal{G}) = 0. \tag{19}$$

**Codimension-$k$ Intersections and Sequential Cuts**

Let us finally turn to the analysis of the structure of higher codimension faces of $\mathcal{P}_{\mathcal{G}}$ – see Figure 4. We will be interested in those intersections of $k > 2$ facets of $\mathcal{P}_{\mathcal{G}}$ that occur *outside* $\mathcal{P}_{\mathcal{G}}$ or, which is the same, that can occur in codimension higher than $k$ as a codimension-$k$ intersection of the facets outside the polytope can be projected onto the polytope in higher codimensions. Consequently, given a set of facets which intersect each other outside our polytope, their intersection with another hyperplane containing a further facet can lie on the polytope and, hence, corresponds to a non-vanishing multiple residue for the canonical form[5]. This is extremely important for the understanding the full face structure of $\mathcal{P}_{\mathcal{G}}$. However our interest is only on the intersections of the facets of $\mathcal{P}_{\mathcal{G}}$ outside $\mathcal{P}_{\mathcal{G}}$.

If $\mathcal{W}^{(\mathfrak{g}_j)}$ is the hyperplane identified by the subgraph $\mathfrak{g}_j \subseteq \mathcal{G}$ and $\mathcal{W}^{(\mathfrak{g}_1 \cdots \mathfrak{g}_k)} := \mathcal{W}^{(\mathfrak{g}_1)} \cap \mathcal{W}^{(\mathfrak{g}_2)} \cap \ldots \cap \mathcal{W}^{(\mathfrak{g}_k)} \neq \varnothing$ is the lower dimensional hyperplane identified by the intersection among the hyperplanes $\{\mathcal{W}^{(\mathfrak{g}_j)}, \, j = 1, \ldots, k\}$, then we are interested in those hyperplanes $\mathcal{W}^{(\mathfrak{g}_1 \cdots \mathfrak{g}_k)}$ such that $\mathcal{P}_{\mathcal{G}} \cap \mathcal{W}^{(\mathfrak{g}_1 \cdots \mathfrak{g}_k)} = \varnothing$ in codimension-$k$.

A trivial observation is that a face of $\mathcal{P}_{\mathcal{G}}$ can have at most codimension $n_s + n_e - 1$, and thus the intersection among $k > n_s + n_e$ hyperplanes corresponding to any collection of subgraphs $\{\mathfrak{g}_j\}_{j=1}^{k}$ is necessarily empty. Hence, a sufficient but not necessary condition for having empty intersections, is that their codimension $k$ is strictly greater than $n_s + n_e$:

$$\mathcal{P}_{\mathcal{G}} \cap \mathcal{W}^{(\mathfrak{g}_1 \cdots \mathfrak{g}_k)} = \varnothing, \qquad \text{if } k > n_s + n_e. \tag{20}$$

Consequently, if $k > n_s + n_e$, the corresponding multiple residue for the canonical form $\omega(\mathcal{Y}, \mathcal{P}_{\mathcal{G}})$ is necessarily zero:

$$\text{Res}_{\mathcal{W}^{(\mathfrak{g}_1)}} \ldots \text{Res}_{\mathcal{W}^{(\mathfrak{g}_k)}} \omega(\mathcal{Y}, \mathcal{P}_{\mathcal{G}}) = 0, \qquad \text{if } k > n_s + n_e. \tag{21}$$

These conditions are trivial and hence they do not impose constraints on the canonical form. We will therefore be interested in those intersections of $k \leq n_s + n_e$ hyperplanes.

Let us then consider a collection of $k \leq n_s + n_e$ hyperplanes $\{\mathcal{W}^{(\mathfrak{g}_j)}, \, j = 1, \ldots, k\}$ each of which is identifed by a subgraph $\mathfrak{g}_j$ such that they are partially overlapping, *i.e.*

$$\mathfrak{g}_{\sigma(1)} \cap \ldots \cap \mathfrak{g}_{\sigma(j-1)} \cap \bar{\mathfrak{g}}_{\sigma(j)} \cap \ldots \cap \bar{\mathfrak{g}}_{\sigma(k)} \neq \varnothing, \tag{22}$$

for all $j \in [1, k]$, where $\sigma(j) \in \{1, \ldots, k\}$ such that $\sigma(i) \neq \sigma(j)$ for all $i, j \in [1, \ldots, k]$ as well as $\sigma(1) < \ldots < \sigma(j-1)$ and $\sigma(j) < \ldots < \sigma(k)$[6]. Notice that all the intersections in (22) but the one containing just the complementary subgraphs $\bar{\mathfrak{g}}_j$ identify lower dimensional scattering facets. The number of such intersections is $2^k - 1$. Furthermore the vertices associated to $\mathfrak{g}_c := (\bar{\mathfrak{g}}_1 \cap \ldots \cap \bar{\mathfrak{g}}_k) \cup \bar{\bar{\mathcal{E}}}$ given by the union of the intersection of all the complementary subgraphs with the cut edges departing from it, identify a polytope $\mathcal{P}_{\mathfrak{g}_c}$ with affine dimension given by $n_s^{\bar{\mathfrak{g}}} + n_e^{(\bar{\mathfrak{g}})} + n_{\bar{\bar{\mathcal{E}}}}$, where $n_s^{(\bar{\mathfrak{g}})}$ and $n_e^{(\bar{\mathfrak{g}})}$ are respectively the number of sites and edges in $\mathfrak{g}_c$ while $n_{\bar{\bar{\mathcal{E}}}}$ is the number of cut edges $\bar{\bar{\mathcal{E}}}$. The dimension of the intersection $\mathcal{P}_{\mathcal{G}} \cap \mathcal{W}^{(\mathfrak{g}_1 \cdots \mathfrak{g}_k)}$ is then given by

$$\dim(\mathcal{P}_{\mathcal{G}} \cap \mathcal{W}^{(\mathfrak{g}_1 \cdots \mathfrak{g}_k)}) = \sum_{\mathcal{S}_{\mathfrak{g}}} (n_s^{(\mathfrak{g})} + n_e^{(\mathfrak{g})} - 1) + n_{\bar{\mathcal{E}}} + n_s^{\bar{\mathfrak{g}}} + n_e^{(\bar{\mathfrak{g}})} + n_{\bar{\bar{\mathcal{E}}}} - 1$$

$$= n_s + n_e - 1 - \sum_{\mathcal{S}_{\mathfrak{g}}} 1, \tag{23}$$

where the sum runs over those intersections among graphs which identify lower dimensional scattering facets. In order for the intersection $\mathcal{P}_{\mathcal{G}} \cap \mathcal{W}^{(\mathfrak{g}_1 \cdots \mathfrak{g}_k)}$ to be of the expected codimension,

---

[5]We would like to thank Lukas Kühne and Leonid Monin for discussions about this point.

[6]This condition is required in order to avoid double counting.

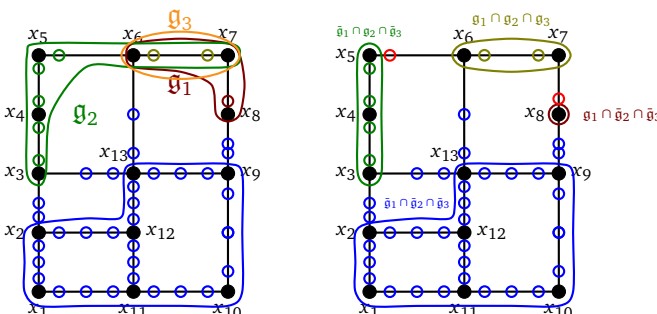

Figure 4: The intersection of the three facets identified by the subgraphs $\mathfrak{g}_1$, $\mathfrak{g}_2$ and $\mathfrak{g}_3$. This codimension-3 face has the same vertex structure as the intersection identified by the partially overlapping graphs $\mathfrak{g}_1$ and $\mathfrak{g}_2$. It factorises into three lower-dimensional scattering facets $\mathcal{S}_{\mathfrak{g}_1 \cap \mathfrak{g}_2 \cap \mathfrak{g}_3}$, $\mathcal{S}_{\mathfrak{g}_1 \cap \bar{\mathfrak{g}}_2 \cap \bar{\mathfrak{g}}_3}$, and $\mathcal{S}_{\bar{\mathfrak{g}}_1 \cap \mathfrak{g}_2 \cap \bar{\mathfrak{g}}_3}$, whose vertices are, respectively, depicted by the markings o, o, and o. The remaining vertices, denoted by o and o, respectively, identify the simplex $\Sigma_{\not{\mathfrak{g}}}$ and $\mathcal{P}_{\mathfrak{g}_c}$ [34].

the sum $\sum_{\mathcal{S}_{\mathfrak{g}}} 1 = 2^k - 1$ should be equal to $k - 1$, *i.e.* $2^k = k$. However, there is no value of $k$ that satisfies this equation, and hence a set of $k$ subgraphs which is mutually partially overlapping never identifies a face of $\mathcal{P}_{\mathcal{G}}$ of codimension $k$. Nevertheless, the configuration of vertices identified by a given set of $k$ mutually partially overlapping graphs can belong to a codimension $2^k$ face which correspond to consider further $2^k - k$ graphs such that no new lower-dimensional scattering facet is generated, *i.e.* each of the new subgraphs should coincide itself with one of the intersections (22).

Thus, in order to have a general condition for a subset of facets to intersect outside of $\mathcal{P}_{\mathcal{G}}$, let us closely analyse the general formula for the dimension of the intersection $\mathcal{P}_{\mathcal{G}} \cap \mathcal{W}^{(\mathfrak{g}_1 \cdots \mathfrak{g}_k)}$:

$$\dim(\mathcal{P}_{\mathcal{G}} \cap \mathcal{W}^{(\mathfrak{g}_1 \cdots \mathfrak{g}_k)}) = \sum_{\mathcal{S}_{\mathfrak{g}}} (n_s^{(\mathfrak{g})} + n_e^{(\mathfrak{g})} - 1) + n_{\not{\mathfrak{g}}} + n_s^{(\mathfrak{g}_c)} + n_e^{(\mathfrak{g}_c)} + n_{\bar{\mathfrak{g}}} - 1, \tag{24}$$

where, as before, the sum runs over the lower-dimensional scattering facets identified by the *non-empty* graphs intersections among (22). In order for the intersection to be the empty set on $\mathcal{P}_{\mathcal{G}}$ in codimension $k$, the dimension (24) has to be strictly less than $n_s + n_e - 1 - k$. Importantly, while the sum over the number of sites of all subsets is always equal to the number $n_s$ of $\mathcal{G}$, the same is not true for the number of edges, as on the intersection $\mathcal{P}_{\mathcal{G}} \cap \mathcal{W}^{(\mathfrak{g}_1 \cdots \mathfrak{g}_k)}$ there might not be any vertex of $\mathcal{P}_{\mathcal{G}}$ attached to some of the cut edges $\not{\mathfrak{g}}$. Let $\not{n}_{\not{\mathfrak{g}}}$ be the number of edges of $\mathcal{G}$ with no vertex on this intersection, then the necessary and sufficient condition for the intersection $\mathcal{W}^{(\mathfrak{g}_1 \cdots \mathfrak{g}_k)}$ to occur *outside* $\mathcal{P}_{\mathcal{G}}$ is

$$\sum_{\mathcal{S}_{\mathfrak{g}}} 1 + \not{n}_{\not{\mathfrak{g}}} > k, \tag{25}$$

where the sum runs over the set of lower-dimensional scattering facets. The condition (25) allows us to construct the codimension-$k$ outer intersections of the facets from the $(k-1)$-codimension ones. Just as an example, let us consider two subgraphs $\mathfrak{g}_1, \mathfrak{g}_2 \subseteq \mathcal{G}$. From our previous discussion, the related hyperplane intersects outside $\mathcal{P}_{\mathcal{G}}$ either if they are partially overlapping or if $\mathfrak{g}_j \subset \mathfrak{g}_i$ such that $n_{\mathfrak{g}_j} > L_{\mathfrak{g}_i} + 1$. All the other possible configurations give rise to non-empty intersections on the polytope. Let us now identify those hyperplanes containing the facets of the polytopes and intersection each other outside it in a codimension-3 subspace, by adding a third subgraph $\mathfrak{g}_3$ to a given pair $(\mathfrak{g}_1, \mathfrak{g}_2)$. If such a pair is already in a configuration such that the related hyperplanes intersect each other outside $\mathcal{P}_{\mathcal{G}}$, then the condition (25) for

$k = 3$ is satisfied if and only if

$$\mathfrak{g}_3 \neq \{\mathfrak{g}_1 \cap \mathfrak{g}_2, \mathfrak{g}_1 \cap \bar{\mathfrak{g}}_2, \bar{\mathfrak{g}}_1 \cap \mathfrak{g}_2\}. \tag{26}$$

If, instead, $\mathfrak{g}_1$ and $\mathfrak{g}_2$ are disjoint and do not have any cut edge departing from them in common, then one can choose $\mathfrak{g}_3$ such that

- it partially overlaps with either one of the $\mathfrak{g}_j$'s or both;

- $\mathfrak{g}_3 \subset \mathfrak{g}_j$, in such a way that $n_{\mathfrak{g}_3} > L_{\mathfrak{g}_j} + 1$;

- $\mathfrak{g}_3 \supset \mathfrak{g}_j$ such that $n_{\mathfrak{g}_j} > L_{\mathfrak{g}_3} + 1$.

If $\mathfrak{g}_1$ and $\mathfrak{g}_2$ were to be disjoint and with some cut edge departing from them in common, then $\mathfrak{g}_3$ can be chosen in such a way that it contains both $\mathfrak{g}_1$ and $\mathfrak{g}_2$ as well as at least one of the cut edge departing from them, *i.e.* $\not\!\eta_{\bar{\mathscr{E}}} \neq 0$.

Finally, we can consider $\mathfrak{g}_1$ and $\mathfrak{g}_2$ so that one is a subgraph of the other one (namely, $\mathfrak{g}_2 \subset \mathfrak{g}_1$) and such that the related facet intersect on the polytope in codimension-2, then $\mathfrak{g}_3$ can be chosen in such way that:

- it partially overlaps with either one of the $\mathfrak{g}_j$'s or both;

- $\mathfrak{g}_3 \subset \mathfrak{g}_j$ with $n_{\mathfrak{g}_3} > L_{\mathfrak{g}_j} + 1$;

- $\mathfrak{g}_3 \supset \mathfrak{g}_j$ with $n_{\mathfrak{g}_j} > L_{\mathfrak{g}_3} + 1$;

- $\mathfrak{g}_3 \subset \mathfrak{g}_1$ and $\mathfrak{g}_2 \cap \mathfrak{g}_3 = \varnothing$ with at least one cut edge departing from $\mathfrak{g}_2$ and $\mathfrak{g}_3$ in common.

In all these cases the triple sequential cut of the canonical form vanishes:

$$\text{Res}_{\mathcal{W}(\mathfrak{g}_1)} \text{Res}_{\mathcal{W}(\mathfrak{g}_2)} \text{Res}_{\mathcal{W}(\mathfrak{g}_3)} \omega(\mathcal{Y}, \mathcal{P}_{\mathcal{G}}) = 0. \tag{27}$$

Notice that if $\mathfrak{g}_1 = \mathcal{G}$ and $\mathfrak{g}_2 \cap \mathfrak{g}_3 \neq \varnothing$ in such a way they share at least one of the cut edges departing from them, the constrain (27) becomes just the statement that the energy in the cut edges of the scattering amplitudes must have a directed flow.

As we stressed earlier, knowing the intersections of the facets outside the polytope can allow us to determine the zeroes of the canonical form and, hence, its numerator. Given a cosmological polytope $\mathcal{P}_{\mathcal{G}} \subset \mathbb{P}^{n_s + n_e - 1}$ with $\tilde{\nu}$ number of facets, then the canonical form can be generically written as

$$\omega(\mathcal{Y}, \mathcal{P}_{\mathcal{G}}) = \frac{\mathfrak{n}_\delta(\mathcal{Y})}{\mathfrak{d}_{\tilde{\nu}}(\mathcal{Y})} \langle \mathcal{Y} d^{n_s + n_e - 1} \mathcal{Y} \rangle, \tag{28}$$

where $\mathfrak{n}_\delta(\mathcal{Y})$ and $\mathfrak{d}_{\tilde{\nu}}(\mathcal{Y})$ are polynomials of degree $\delta := \tilde{\nu} - n_s - n_e$ and $\tilde{\nu}$ respectively. In particular, the numerator $\mathfrak{n}_\delta(\mathcal{Y})$ is determined by a totally symmetric $\delta$-tensor — $\mathfrak{n}_\delta := \mathcal{C}_{I_1 \dots I_\delta} \mathcal{Y}^{I_1} \dots \mathcal{Y}^{I_\delta}$ — whose number $\Delta$ of degrees of freedom is given by

$$\Delta = \binom{n_s + n_e + \delta - 1}{\delta} - 1, \tag{29}$$

and the symmetric tensor $\mathcal{C}_{I_1 \dots I_\delta}$ precisely parametrises the locus of the intersections of the facets outside $\mathcal{P}_{\mathcal{G}}$ and it is determined by the vanishing multiple-residue conditions just discussed. Importantly, fixing $\mathcal{C}$ via such conditions can be a non-trivial task and can be explicitly and straightforwardly performed in simple enough cases (see the Appendix). Nevertheless, the conditions on the multiple-residues can allow us to systematically construct signed triangulations which involve subspaces of the locus $\mathcal{C}$ and, consequently, various ways of determining the canonical form of $\mathcal{P}_{\mathcal{G}}$ as sum of the canonical forms of the polytopes which

signed-triangulate it.

**Outer Intersections and Triangulations**

Our discussion will be focused on the class of signed-triangulations which involve just a single subspace of the locus $\mathcal{C}$, *i.e.* all the elements of the collection of polytopes $\{\mathcal{P}_{\mathcal{G}}^{(j)}\}$ triangulating $\mathcal{P}_{\mathcal{G}}$ share just one higher codimension face. In general, such a collection of polytopes provides a polytope subdivision rather than a (signed) triangulation, *i.e.* not necessarily all the elements of the collection are simplices. Thus, in order to obtained an actual signed triangulation, we need some extra conditions on the higher codimension face. Once we have identified which subspaces of the locus $\mathcal{C}$ can be used for triangulating $\mathcal{P}_{\mathcal{G}}$ through them, then we can immediately identify the collection of simplices for the triangulation through one of such subspaces, and write down the canonical form $\omega(\mathcal{Y}.\mathcal{P}_{\mathcal{G}})$ using the multiple residue conditions discussed in the previous section. For the sake of clarity, let us first discuss how to write down the canonical form in terms of such triangulations, and then how to find the subspaces of the locus $\mathcal{C}$ which allow for such triangulations.

Let $q_{\mathfrak{g}}(\mathcal{Y}) := \mathcal{W}_I^{(\mathfrak{g})} \mathcal{Y}^I$, where, as usual, $\mathcal{W}_I^{(\mathfrak{g})}$ is the hyperplane identified by the subgraph $\mathfrak{g} \subseteq \mathcal{G}$ and such that $\mathcal{P}_{\mathcal{G}} \cap \mathcal{W}^{(\mathfrak{g})} \neq \varnothing$ is a facet of $\mathcal{P}_{\mathcal{G}}$. Then, the canonical form $\omega(\mathcal{Y}, \mathcal{P}_{\mathcal{G}})$ associated to $\mathcal{P}_{\mathcal{G}}$ can be generically written as

$$\omega(\mathcal{Y}, \mathcal{P}_{\mathcal{G}}) = \frac{\mathfrak{n}_\delta(\mathcal{Y})}{\displaystyle\prod_{\mathfrak{g} \subseteq \mathcal{G}} q_{\mathfrak{g}}(\mathcal{Y})} \langle \mathcal{Y} d^{n_s+n_e-1} \mathcal{Y} \rangle , \tag{30}$$

where $\delta = \tilde{\nu} - n_s - n_e$ and $\tilde{\nu}$ is the number of facets of $\mathcal{P}_{\mathcal{G}}$.

Let $\mathfrak{G}_\circ := \{\mathfrak{g}_j\}_{j=1}^k$ be the set of subgraphs which identify the $k$-dimensional hyperplane $\mathcal{W}^{(\mathfrak{g}_1 \cdots \mathfrak{g}_k)}$ such that $\mathcal{P}_{\mathcal{G}} \cap \mathcal{W}^{(\mathfrak{g}_1 \cdots \mathfrak{g}_k)} = \varnothing$ and identifies the codimension-$k$ subspace of $\mathcal{C}$ through which we want to triangulate $\mathcal{P}_{\mathcal{G}}$. Then, each simplex of the triangulation we are looking for is identified by the inequalities $\{q_{\mathfrak{g}_j} \geq 0\}_{j=1}^k$ associated to the codimension-$k$ intersection outside $\mathcal{P}_{\mathcal{G}}$ in question as well as further $n_s + n_e - k$ inequalities $\{q_{\mathfrak{g}} \geq 0\}_{g \notin \mathcal{G}_\circ}$ such that the hyperplanes associated to the linear polynomials $\{q_{\mathfrak{g}}\}_{\mathfrak{g} \notin \mathfrak{G}_\circ}$ have non-vanishing codimension-$(n_s + n_e - k)$ on $\mathcal{P}_{\mathcal{G}}$:

$$\mathrm{Res}_{\mathcal{W}^{(\mathfrak{g}_{\sigma(1)})}} \dots \mathrm{Res}_{\mathcal{W}^{(\mathfrak{g}_{\sigma(n_s+n_e-k)})}} \omega(\mathcal{Y}, \mathcal{P}_{\mathcal{G}}) \neq 0 . \tag{31}$$

The collection of simplices signed-triangulating $\mathcal{P}_{\mathcal{G}}$ is therefore given by the possible collections of $(n_s+n_e-k)$ subgraphs in $\mathcal{G}_0$ such that (31) holds. Consequently, the canonical form $\omega(\mathcal{Y}, \mathcal{P}_{\mathcal{G}})$ can be written as

$$\omega(\mathcal{Y}, \mathcal{P}_{\mathcal{G}}) = \sum_{\sigma \in \mathfrak{G}_k} \prod_{l=1}^{n_s+n_e-k} \frac{1}{q_{\sigma(l)}(\mathcal{Y})} \frac{\langle \mathcal{Y} d^{n_s+n_e-1} \mathcal{Y} \rangle}{\displaystyle\prod_{j=1}^k q_{\mathfrak{g}_j}(\mathcal{Y})} , \tag{32}$$

where $\mathfrak{G}_k$ is the collection of sets of $n_s + n_e - k$ subgraphs not involving any element of $\mathfrak{G}_\circ$ and identifying codimension-$(n_s + n_e - k)$ faces of $\mathcal{P}_{\mathcal{G}}$. Importantly, formula (32) provides a general expression of the canonical function for *any* triangulation through a specific class of subspace of the locus $\mathcal{C}$ and it contains just physical poles at $q_{\mathfrak{g}}(\mathcal{Y}) := \mathcal{W}_I^{(\mathfrak{g})} \mathcal{Y}^I = 0$ for all $\mathfrak{g} \subseteq \mathcal{G}$. We need to characterise the subspaces of the locus $\mathcal{C}$ which allow for such triangulations (and not just for polytope subdivisions).

In the previous section we showed how the vertex structure of the intersection of $k$ facets identified by the subgraphs $\{\mathfrak{g}_j\}_{j=1}^k$ determines whether it lies on $\mathcal{P}_{\mathcal{G}}$, constituting one of its codimension-$k$ faces, or outside $\mathcal{P}_{\mathcal{G}}$, identifying a codimension-$k$ subspace of the locus $\mathcal{C}$ of the zeroes of the canonical form $\omega(\mathcal{Y}, \mathcal{P}_{\mathcal{G}})$. Recall that given a subgraph $\mathfrak{g}$, the vertex structure of

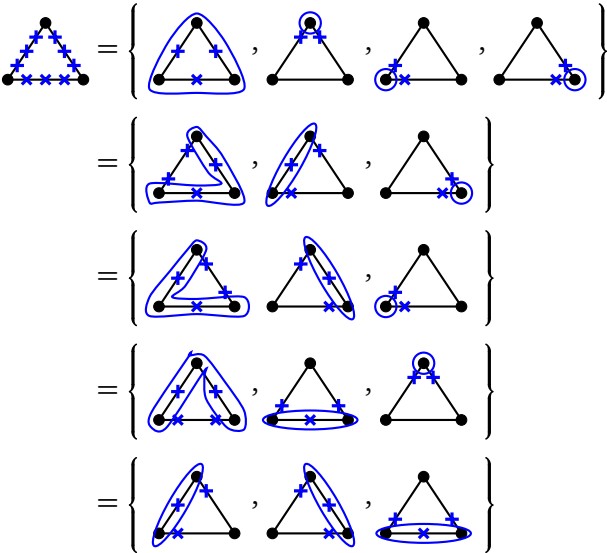

Figure 5: Higher codimension intersections of the facets outside $\mathcal{P}_{\mathcal{G}}$. They can be graphically obtained by considering a set of $k < n_s + n_e$ subgraphs which completely marks the graph $\mathcal{G}$ in such a way that they do not share any marking.

the facet $\mathcal{P}_{\mathcal{G}} \cap \mathcal{W}^{(\mathfrak{g})}$ can be obtained graphically by marking with ✗ all the internal edges of $\mathfrak{g}$ in the middle, while all the edges departing from it close to the sites in $\mathfrak{g}$. Then, intersections of the facets outside $\mathcal{P}_{\mathcal{G}}$ in a higher codimension subspace can be obtained by considering a set of $k < n_s + n_e$ subgraphs $\{\mathfrak{g}_j\}_{j=1}^k$ such that the graph $\mathcal{G}$ is completely marked. A special class of such sets is constituted by those whose elements introduce complementary markings, *i.e.* the subgraphs of a given set do not share any marking (see Figure 5).

Let $\mathfrak{G}_\circ$ and $\mathfrak{G}'_\circ$ be two of such special sets of subgraphs. Then, the fact that both of them mark completely the graph $\mathcal{G}$, implies an existence of a linear relation among the hyperplanes identified by the elements of the two sets:

$$\sum_{\mathfrak{g} \in \mathfrak{G}_\circ} \mathcal{W}_I^{(\mathfrak{g})} \sim \sum_{\mathfrak{g}' \in \mathfrak{G}'_\circ} \mathcal{W}_I^{(\mathfrak{g}')}, \tag{33}$$

where "$\sim$" just indicates that it is a projective relation.

Let us now consider $\mathfrak{G}_\circ$ and a set $\mathfrak{G}_c := \{\mathfrak{g}_j\}$ of $n_s + n_e - k$ subgraphs that *are not* contained in $\mathfrak{G}_\circ$ and identifies a codimension-$(n_s + n_e - k)$ face of $\mathcal{P}_{\mathcal{G}}$. Notice that as we go on such faces, the intersection identified by $\mathfrak{G}_\circ$ gets projected onto them and hence define one of the highest codimension boundaries: defining $\mathcal{W}^{(\mathfrak{G}_\circ)} := \bigcap_{\mathfrak{g} \in \mathfrak{G}_\circ} \mathcal{W}^{(\mathfrak{g})}$ and $\mathcal{W}^{(\mathfrak{G}_c)} := \bigcap_{\mathfrak{g} \in \mathfrak{G}_c} \mathcal{W}^{(\mathfrak{g})}$, then $\mathcal{P}_{\mathcal{G}} \cap \mathcal{W}^{(\mathfrak{G}_\circ)} \cap \mathcal{W}^{(\mathfrak{G}_\chi)} \subseteq \mathbb{P}^0$, then it is not vanishing and its canonical form is a number. This precisely implies that $\mathfrak{G}_\circ$ and any set $\mathfrak{G}_c$ of $n_s + n_e - k$ subgraphs that *are not* contained in $\mathfrak{G}_\circ$ and identifies a codimension-$(n_s + n_e - k)$ face of $\mathcal{P}_{\mathcal{G}}$, define a simplex in $\mathbb{P}^{n_s+n_e-1}$. Finally, summing over all these simplices for a given $\mathfrak{G}_\circ$ cover the full cosmological polytope $\mathcal{P}_{\mathcal{G}}$. Hence, we can write:

$$\omega(\mathcal{Y}, \mathcal{P}_{\mathcal{G}}) = \sum_{\{\mathfrak{G}_c\}} \prod_{\mathfrak{g}' \in \mathfrak{G}_c} \frac{1}{q_{\mathfrak{g}'}(\mathcal{Y})} \frac{\langle \mathcal{Y} d^{n_s+n_e-1} \mathcal{Y} \rangle}{\prod\limits_{\mathfrak{g} \in \mathfrak{G}_0} q_{\mathfrak{g}}(\mathcal{Y})}. \tag{34}$$

It is interesting to notice that for any graph $\mathcal{G}$, one of the sets of subgraphs completely marking $\mathcal{G}$ and such that its elements introduce complementary markings, is constituted by the graph $\mathcal{G}$ and all the subgraphs $\mathfrak{g}_s$ containing a single site $s$. They identify a codimension-$(n_s + 1)$ subspace of the locus $\mathcal{C}$ of the intersections of the facets of $\mathcal{P}_{\mathcal{G}}$ outside $\mathcal{P}_{\mathcal{G}}$. Hence,

expression (34) for the canonical form considering $\mathfrak{G}_\circ := \{\mathcal{G}, \{\mathfrak{g}_s\}_{s\in\mathcal{V}}\}$ then becomes

$$
\omega(\mathcal{Y}, \mathcal{P}_\mathcal{G}) = \sum_{\{\mathfrak{G}_c\}} \prod_{\mathfrak{g}'\in\mathfrak{G}_c} \frac{1}{q_{\mathfrak{g}'}(\mathcal{Y})} \frac{\langle \mathcal{Y} d^{n_s+n_e-1} \mathcal{Y} \rangle}{q_\mathcal{G}(\mathcal{Y}) \prod_{s\in\mathcal{V}} q_{\mathfrak{g}_s}(\mathcal{Y})} .
\tag{35}
$$

Notice that, together with the $(n_s+1)$ boundaries related to the subgraphs in $\mathfrak{G}_\circ$, each simplex in (35) has other $n_e - 1$ boundaries. The sum over the sets of facets whose intersection is on $\mathcal{P}_\mathcal{G}$ then corresponds to recursively erase an edge in the graph: this is precisely the OFPT recursion relation proven in [35]!

All the other possible choices for $\mathfrak{G}_\circ$ provide novel representations for the wavefunction of the universe. Explicit examples are provided in the Appendix. It is important to emphasise that all the representations obtained in this way not only are characterised by having just physical poles, but also they make manifest a subset of the compatible channels and non-compatible channels, *i.e.* a subset of non-vanishing and vanishing multiple residues respectively. Said differently, the representations of the wavefunctions that can be obtained as a triangulation of the cosmological polytope through a subspace of the locus $\mathcal{C}$, have the inherent feature of making manifest a subset of the Steinmann-like relations and their generalisation to higher codimension singularities.

## 4  Representations for flat-space amplitudes

The very same discussion we carried out in the previous section for the cosmological polytope $\mathcal{P}_\mathcal{G}$ can be performed for its scattering facet $\mathcal{S}_\mathcal{G}$. Namely, we would like to know which intersections $\mathcal{S}_\mathcal{G} \cap \mathcal{W}^{(\mathfrak{g}_1\cdots\mathfrak{g}_k)}$ are empty in codimension-$k$ and hence which hyperplane $\mathcal{W}^{(\mathfrak{g}_1\cdots\mathfrak{g}_k)}$ lies outside of $\mathcal{S}_\mathcal{G}$ and defines a codimension-$k$ subspace of the locus of the zeroes of the canonical form of $\mathcal{S}_\mathcal{G}$ and identify which of them allows for triangulations of $\mathcal{S}_\mathcal{G}$.

First, given a graph $\mathcal{G}$ with $L$ loops, the analysis of whether $\mathcal{S}_\mathcal{G} \cap \mathcal{W}^{(\mathfrak{g}_1\cdots\mathfrak{g}_k)}$ is empty or not, involves all those subgraphs $\mathfrak{g}_j \subset \mathcal{G}$ such that the number $n_{\mathfrak{g}_j}$ departing from it is greater than $L+1$: for $n_{\mathfrak{g}_j} \in [1, L+1]$ then $\mathcal{S}_\mathcal{G} \cap \mathcal{W}^{(\mathfrak{g}_i)} = \varnothing$ and the subgraph $\mathfrak{g}_j$ does not identify a singularity of the canonical form of the scattering facet. With this condition in mind for a subgraph to identify a boundary of $\mathcal{S}_\mathcal{G}$, whether $\mathcal{S}_\mathcal{G} \cap \mathcal{W}^{(\mathfrak{g}_1\cdots\mathfrak{g}_k)}$ is empty or not in codimension-$k$ is again determined by the dimension counting of the lower-dimensional polytopes which $S_\mathcal{G} \cap \mathcal{W}^{(\mathfrak{g}_1\cdots\mathfrak{g}_k)}$ factorises into:

$$
\dim(S_\mathcal{G} \cap \mathcal{W}^{(\mathfrak{g}_1\cdots\mathfrak{g}_k)}) = \sum_{\mathcal{S}_\mathfrak{g}} (n_s^{(\mathfrak{g})} + n_e^{(\mathfrak{g})} - 1) + n_{\not{\mathcal{E}}} - 1 ,
\tag{36}
$$

where the sum runs over the lower-dimensional scattering facets identified by the intersections among the graphs $\mathfrak{g}_j$ and their complementary graphs $\bar{\mathfrak{g}}_j$. In order for this intersection to be empty, its dimension (36) should be strictly less than $n_s + n_e - 2 - k$. If $\not{n}_{\not{\mathcal{E}}}$ is the number of edges of $\mathcal{G}$ with no vertex on this intersection attached to it, then the condition for having $S_\mathcal{G} \cap \mathcal{W}^{(\mathfrak{g}_1\cdots\mathfrak{g}_k)} = \varnothing$ in codimension $k$, is given by

$$
\sum_{\mathcal{S}_\mathfrak{g}} 1 + \not{n}_{\not{\mathcal{E}}} > k+1 .
\tag{37}
$$

This condition allows us to identify all those intersections among the hyperplanes containing the facets of $\mathcal{S}$, that lie outside $\mathcal{S}_\mathcal{G}$ on the locus $\mathcal{C}(\mathcal{S}_\mathcal{G})$ of the zeroes of the canonical form $\omega(\mathcal{Y}, \mathcal{S}_\mathcal{G})$ and hence the vanishing multiple residue conditions the latter has to satisfy

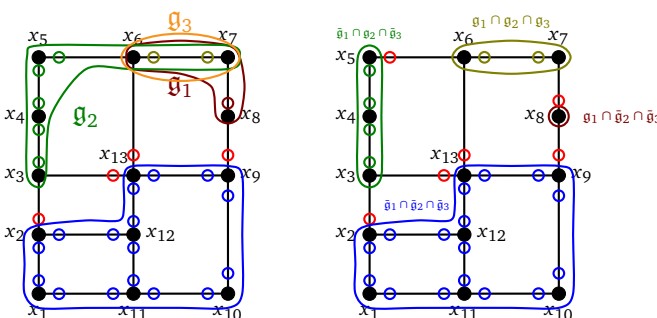

Figure 6: The intersection of the three facets identified by the subgraphs $\mathfrak{g}_1$, $\mathfrak{g}_2$ and $\mathfrak{g}_3$ on the scattering facet $\mathcal{S}_\mathcal{G}$ [34].

As for the full cosmological polytope case, we are interested to identify those subspaces of $\mathcal{C}(\mathcal{S}_\mathcal{G})$ through each of which we can triangulate the scattering facet. Again, this special class of intersections can be identified by the sets of $k < n_s + n_e - 1$ subgraphs of $\mathcal{G}$ that completely marks $\mathcal{G}$ in such a way that the subgraphs in a given set do not share any marking.

Let $\mathfrak{G}_\circ$ and $\mathfrak{G}_c$ be respectively any of those sets and a set of $n_s + n_e - k - 1$ subgraphs which do not belong to $\mathfrak{G}_\circ$ and that identify a codimension-$(n_s + n_e - k - 1)$ face of $\mathcal{S}_\mathcal{G}$. If $\mathcal{W}^{(\mathfrak{G}_\circ)} := \bigcap_{\mathfrak{g} \in \mathfrak{G}_\circ} \mathcal{W}^{(\mathfrak{g})}$ and $\mathcal{W}^{(\mathfrak{G}_c)} := \bigcap_{\mathfrak{g} \in \mathfrak{G}_c} \mathcal{W}^{(\mathfrak{g})}$, then $\mathcal{S}_\mathcal{G} \cap \mathcal{W}^{(\mathfrak{G}_\circ)} \cap \mathcal{W}^{(\mathfrak{G}_c)} \subset \mathbb{P}^0$, it is not empty and its canonical form is a constant. Then the canonical form for the scattering facet can be written as

$$\omega(\mathcal{Y}, \mathcal{S}_\mathcal{G}) = \sum_{\{\mathfrak{G}_c\}} \prod_{\mathfrak{g}' \in \mathfrak{G}_c} \frac{1}{q_{\mathfrak{g}'}(\mathcal{Y})} \frac{\langle \mathcal{Y} d^{n_s + n_e - 2} \mathcal{Y} \rangle}{\prod_{\mathfrak{g} \in \mathfrak{G}_0} q_\mathfrak{g}(\mathcal{Y})}. \tag{38}$$

These signed-triangulations through a single subspace of the locus $\mathcal{C}(\mathcal{S}_\mathcal{G})$ of the zeroes of the canonical form of the scattering facet $\mathcal{S}_\mathcal{G}$ provide several representations for scattering amplitudes, some of which are novel to our knowledge. Two of them can be written in a general fashion, irrespectively of the topology of the graph $\mathcal{G}$.

The first one can be obtained by considering $\mathfrak{G}_\circ$ as the set of all the subgraphs $g_s$ defined by a single site $s$ of $\mathcal{G}$ (see the first line in Figure 7). They identify a subspace of $\mathcal{C}(\mathcal{S}_\mathcal{G})$ of dimension $n_s$. Then, any $\mathfrak{G}_c$ defines an $(n_e - 1)$-dimensional face of $\mathcal{G}_c$: the sum over $\{\mathfrak{G}_c\}$ corresponds to recursively erase an edge in the graph. This is the OFPT representation for amplitudes:

$$\omega(\mathcal{Y}, \mathcal{S}_\mathcal{G}) = \sum_{\{\mathfrak{G}_c\}} \prod_{\mathfrak{g}' \in \mathfrak{G}_c} \frac{1}{q_{\mathfrak{g}'}(\mathcal{Y})} \frac{\langle \mathcal{Y} d^{n_s + n_e - 2} \mathcal{Y} \rangle}{\prod_{s \in \mathcal{V}} q_{\mathfrak{g}_s}(\mathcal{Y})}. \tag{39}$$

The second one can be obtained by considering $\mathfrak{G}_\circ$ as the set of all the subgraphs $\mathfrak{g}_e$ containing all the sites of $\mathcal{G}$ as well as all its edges but one, which we label with $e$ (see the last line in Figure 7). They identify a subspace of $\mathcal{C}(\mathcal{S}_\mathcal{G})$ of dimension $n_e$. Then, any $\mathfrak{G}_c$ defines an $(n_s - 1)$-dimensional face of $\mathcal{S}_\mathcal{G}$ and the canonical form can be written as

$$\omega(\mathcal{Y}, \mathcal{S}_\mathcal{G}) = \sum_{\{\mathfrak{G}_c\}} \prod_{\mathfrak{g}' \in \mathfrak{G}_c} \frac{1}{q_{\mathfrak{g}'}(\mathcal{Y})} \frac{\langle \mathcal{Y} d^{n_s + n_e - 2} \mathcal{Y} \rangle}{\prod_{e \in \mathcal{E}} q_{\mathfrak{g}_e}(\mathcal{Y})}. \tag{40}$$

Importantly, $q_{\mathfrak{g}_e}(\mathcal{Y}) := \mathcal{W}_I^{(\mathfrak{g}_e)} \mathcal{Y}^I = 2y_e$, where $y_e$ is the label associated at the edge that $\mathfrak{g}_e$ cuts (i.e. it is the energy associated to the cut edge) and the prefactor $\prod_{e \in \mathcal{E}} (2y_e)^{-1}$ constitutes the Lorentz-invariant phase-space measure, and the sum is over the set of compatible channels making manifest the Steinmann relations. This is precisely the causal representation conjectured in [25]! The proof of the existence of the triangulations of the scattering facet through

the subspace of $\mathcal{C}(\mathcal{S}_{\mathcal{G}})$ identified by the set $\mathfrak{G}_{\circ}$ (38) as well as the possibility of choosing $\mathfrak{G}_{\circ}$ in such a way to provide the Lorentz-invariant phase-space measure, provides a general (combinatorial) proof of the causal representation. Furthermore, the fact that $\mathfrak{G}_c$ has dimension $n_s - 1$ implies that the canonical form, and consequently the Feynman graph contribution to the scattering amplitude, has the very same structure for all the graphs with the same number of sites but different number of edges (they can be obtained from a given graph by just adding edges between two sites): the prefactor related to the Lorentz-invariant phase-space increases in dimension, while the structure of compatible channels stays invariant. This feature was first observed in [25], and (40) provides a proof of it.

## 5 Conclusion and outlook

Knowing the possible ways of organising perturbation theory is of crucial importance for both extracting fundamental physics out of the relevant observables and having efficient ways of computing them. In this work, we presented a systematic way of generating different representations for both the wavefunction of the universe in cosmology and flat-space scattering amplitudes, exploiting their invariant definition in terms of cosmological polytopes. The distinctive feature of all these representations is to have physical poles only.

We explored the higher codimension singularity structure of both the wavefunction of the universe and the scattering amplitudes, which is transparently encoded in the combinatorial structure of the faces of the cosmological polytope and its scattering facet, respectively. This allowed us to derive further constraints on these observables, restricting their analytic struc-

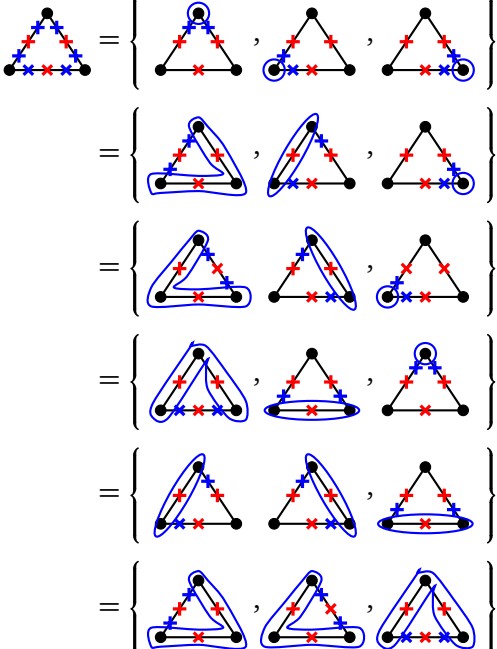

Figure 7: Higher codimension intersections of the facets outside $\mathcal{S}_{\mathcal{G}}$. They can be graphically obtained by considering a set of $k < n_s + n_e$ subgraphs which completely marks the graph $\mathcal{G}$ in such a way that they do not share any marking. The red marking ✗, while the blue one ✗ indicates the one introduced by a subgraph. All but the last are in common with the cosmological polytope analysis — for the full cosmological polytope, the last set can appear in a triangulation through more than one subsets of the locus $\mathcal{C}(\mathcal{P}_{\mathcal{G}})$.

ture. Such constraints extend the Steinmann-like relations. Importantly, the intersections of the facets outside the relevant polytope determine the locus of the zeroes of its canonical form, *i.e.* they determine which multiple discontinuities vanish. We could therefore derive compatibility conditions on the multiple channels and use them to find representations for both the wavefunction of the universe and the scattering amplitude with physical singularities only. In the combinatorial language, such representations are given by all the possible triangulations through the intersections of the facets outside the relevant polytope.

Despite all these representations at first sight look different among each other, they really exploit the very same mechanism: all of them can be seen as being sum of terms each of which has in common a prefactor which identifies one of the zeroes and differ because of the compatibility conditions on the poles which do not appear in the prefactor. Our combinatorial description allows us to consider all these representations on the same footing, recovering both the Old-fashioned perturbation theory (for the wavefunction and the flat-space amplitudes) and the causal representation (just for amplitudes) as well as find novel ones. The presence of the latter suggests that alternative approaches to embark calculations in QFT may allow us to both improve our computational techniques as well as our general understanding of the structure of the physical processes in both flat-space and cosmology.

On the scattering facet, we could provide a proof of the causal representation for scattering amplitudes. Remarkably, this representation was obtained as by-product of the loop-tree duality formalism — a procedure to calculate multi-loop scattering amplitudes — and then conjectured to hold at all orders in perturbation theory. In this paper, the direct connection to one specific triangulation of the scattering facet allowed us to provide a simple, combinatorial, proof of the all-loop conjecture.

An important aspect of our analysis is the possibility to identify a very specific class of triangulations for the cosmological polytopes as well as its scattering facet. In such identification, a crucial role was played by the one-to-one correspondence between our polytopes and graphs as well as the possibility of analysing its face structure via graphical markings. Nevertheless, the fundamental property which has been exploited — *i.e.* the locus of the zeroes of the canonical form is determined by the intersections of the facets outside the polytope — is not specific to the class of polytopes we dealt with but it is indeed a feature of *any* polytope and it is thought to characterise also more general positive geometries. Hence, it would be interesting to see whether a generalisation of the analysis presented in this paper can allow us to tame at least a larger class of signed-triangulations, as well as if it can be extended to other positive geometries of relevance in physics, such as the associahedron [59, 60], the amplituhedra [61] and momentum amplituhedra [62, 63]. The purpose of such an analysis is two-fold: on one hand, finding signed-triangulations, or more generally subdivisions, is not an easy task, they are understood in some specific cases, *e.g.* regular subdivision for cyclic polytopes [64–68], and some more general studies for the amplituhedra started just recently [69]; on the other hand, these other positive geometries encode the full scattering amplitudes rather than an individual graph only, providing us with the possibility of understanding the novel representations we found for both the full observable as well as for integrands with non-trivial numerators. It is worth emphasising that, despite the fact we dealt with scalar integrals only, our results concerning the multiple residues are more general. In effect, our findings can be extended to multi-loop scattering amplitudes, where tensor integrals can yet be analysed by means of relations at integrand and integral level. Therefore, the statements on the individual integrals reflect on the properties of the full amplitude. In the case of the wavefunction of the universe, our results are valid as long as the states involved have a flat-space counterpart.

Finally, based on the analytic structure of the novel representations we found, it would be interesting to carry out numerical studies to elucidate further strengths in the computation of physical observables. Remarkably, given that the various representations evaluate to the same

quantity, we can concentrate on efficiency in their evaluation time, that due to the presence of only physical singularities, gets improved.

## Acknowledgments

We would like to thank Lukas Kühne and Leonid Monin for insightful discussions. Checks have been performed with the aid of `polymake` [70] and `TOPCOM` [71]. The figures have been drawn with TikZ [72]. This research received funding from the European Research Council (ERC) under the European Union's Horizon 2020 research and innovation programme (grant agreement No 725110), *Novel structures in scattering amplitudes*.

## A  Fixing the canonical form in an invariant way

It is instructive to illustrate the idea with the simplest non-trivial example, the three-site line graph

$$
\begin{array}{c}
y_{12} \qquad y_{23} \\
\bullet\!\!-\!\!-\!\!-\!\!\bullet\!\!-\!\!-\!\!-\!\!\bullet \\
x_1 \quad\; x_2 \quad\; x_3
\end{array} ,
$$

whose associated polytope $\mathcal{P}_{\mathcal{G}}$ is the convex hull of six vertices in $\mathbb{P}^4$:

$$
\{\mathbf{x}_1 - \mathbf{y}_{12} + \mathbf{x}_2, \; \mathbf{x}_1 + \mathbf{y}_{12} - \mathbf{x}_2, \; -\mathbf{x}_1 + \mathbf{y}_{12} + \mathbf{x}_2,
$$
$$
\mathbf{x}_2 - \mathbf{y}_{23} + \mathbf{x}_3, \; \mathbf{x}_2 + \mathbf{y}_{23} - \mathbf{x}_3, \; -\mathbf{x}_2 + \mathbf{y}_{23} + \mathbf{x}_3\} .
$$

The codimension-2 intersections of the facets outside $\mathcal{P}_{\mathcal{G}}$ are given by

$$
\tag{41}
$$

and

$$
\tag{42}
$$

which are, respectively, identified by the 3-tensors

$$
\mathcal{Z}_A^{IJK} = \varepsilon^{IJKLM} \mathcal{W}_L^{(\mathcal{G})} \mathcal{W}_M^{(2)} , \tag{43}
$$

and

$$
\mathcal{Z}_B^{IJK} = \varepsilon^{IJKLM} \mathcal{W}_L^{(12)} \mathcal{W}_M^{(23)} , \tag{44}
$$

with $\mathcal{W}_I^{(\mathcal{G})}$, $\mathcal{W}_I^{(2)}$, $\mathcal{W}_I^{(12)}$ and $\mathcal{W}_I^{(23)}$ being the hyperplanes given in order by the set of markings above.

As described earlier, we can easily obtain the codimension-3 intersections from the codimension-2 ones just found by considering an additional hyperplane $s$ such that the condition (25) is satisfied for $k = 3$:

- $\mathcal{Z}_C^{IJ} := \varepsilon^{IJKLM} \mathcal{W}_K^{(\mathcal{G})} \mathcal{W}_L^{(1)} \mathcal{W}_M^{(2)}$

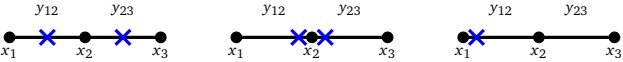

- $\mathcal{Z}_D^{IJ} := \varepsilon^{IJKLM} \mathcal{W}_K^{(\mathcal{G})} \mathcal{W}_L^{(2)} \mathcal{W}_M^{(3)}$

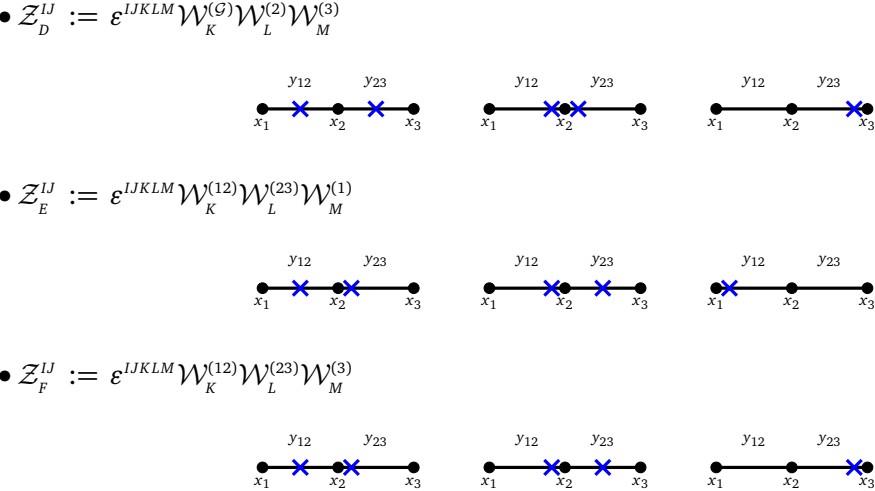

- $\mathcal{Z}_E^{IJ} := \varepsilon^{IJKLM} \mathcal{W}_K^{(12)} \mathcal{W}_L^{(23)} \mathcal{W}_M^{(1)}$

- $\mathcal{Z}_F^{IJ} := \varepsilon^{IJKLM} \mathcal{W}_K^{(12)} \mathcal{W}_L^{(23)} \mathcal{W}_M^{(3)}$

Interestingly, this information is enough to determine the numerator of the canonical form, which in this case is encoded by a co-vector $\mathcal{C}_I$ with just 3 degrees of freedom:

$$\mathcal{C}_I \sim \varepsilon_{IJKLM} \mathcal{Z}_C^{JK} \mathcal{Z}_F^{LM} \sim \varepsilon_{IJKLM} \mathcal{Z}_D^{JK} \mathcal{Z}_E^{LM}, \tag{45}$$

where "$\sim$" indicates that $\mathcal{C}_I$ is determined up-to a numerical coefficient that can be fixed by requiring the invariance of the canonical form under $GL(1)$-transformations on the vectors of the vertices (or, equivalently, of the facets) of $\mathcal{P}_{\mathcal{G}}$.

# B Representations for the wavefunction coefficients: explicit examples

In this section we illustrate, by mean of simple example, how to explicitly compute the triangulations for the cosmological polytope $\mathcal{P}_{\mathcal{G}}$ through a single hyperplane contained in the locus $\mathcal{C}(\mathcal{P}_{\mathcal{G}})$ of the intersections of the facets of $\mathcal{P}_{\mathcal{G}}$ outside $\mathcal{P}_{\mathcal{G}}$. One of them returns the OFPT representations for the wavefunction of the universe, while the others are novel.

**The 1-loop 2-site graph**

Let us begin with the simplest but non-trivial example, the 1-loop 2-site graph. The associated cosmological polytope is a truncated tetrahedron in $\mathbb{P}^3$ (see Figure 2).

The locus $\mathcal{C}(\mathcal{P}_{\mathcal{G}})$ of the intersections of the facets of $\mathcal{P}_{\mathcal{G}}$ outside $\mathcal{P}_{\mathcal{G}}$ is a 2-plane identified by a point (the intersection of its three square facets) and a line (the intersection of its two triangular facets). Such intersections are identified by the sets of subgraphs which completely mark $\mathcal{G}$ and whose marking is complementary (see Figure 8).

We can triangulate $\mathcal{P}_{\mathcal{G}}$ through these intersections, obtaining two different ways of writing its canonical form and, thus, two different representations for the wavefunction with physical poles only. The generic expression for any of these representations is given by (34).

Let us take $\mathfrak{G}_\circ$ to be the set of graphs $\{\mathcal{G}, \mathfrak{g}_1, \mathfrak{g}_2\}$ related to the three square facets, which is identified by the first line in Figure 8 — here we indicate the subgraph constituted by just the site $j$ with $\mathfrak{g}_j$. Then, each $\mathfrak{G}_c$ contains just one of the two subgraphs $\mathfrak{g}_a$ and $\mathfrak{g}_b$ given by the two sites and just one edge: they are precisely the ones contained in the second line in Figure 8, implying that they are mutually incompatible. Hence, the canonical function $\Omega$ can

Figure 8: Intersections of the facets of the cosmological polytope $\mathcal{P}_{\mathcal{G}}$, associated to the 1-loop 2-site graph, outside $\mathcal{P}_{\mathcal{G}}$. In this case, the list above exhausts all the zeroes of the associated canonical form. The fact that the two set of graphs completely mark the graph implies the equivalence relation $\mathcal{W}^{(\mathcal{G})} + \mathcal{W}^{(\mathfrak{g}_1)} + \mathcal{W}^{(\mathfrak{g}_2)} \sim \mathcal{W}^{(\mathfrak{g}_a)} + \mathcal{W}^{(\mathfrak{g}_b)}$ among the hyperplanes identified by the subgraphs.

be written as[7]

$$
\begin{aligned}
\Omega &= \left[ \frac{1}{q_{\mathfrak{g}_a}} + \frac{1}{q_{\mathfrak{g}_b}} \right] \frac{1}{q_{\mathcal{G}} q_{\mathfrak{g}_1} q_{\mathfrak{g}_2}} \\
&= \left[ \frac{1}{x_1 + x_2 + 2y_a} + \frac{1}{x_1 + x_2 + 2y_b} \right] \\
&\quad \times \frac{1}{(x_1 + x_2)(x_1 + y_a + y_b)(x_2 + y_a + y_b)},
\end{aligned}
\tag{46}
$$

where the second equality has been obtained by choosing the local coordinates $\mathcal{Y} = (x_1, y_a, y_b, x_2)$ in $\mathbb{P}^3$, where the $x$'s and $y$'s are label associated to the sites and edges respectively of $\mathcal{G}$. This representation corresponds to the OFPT.

Let us now take $\mathfrak{G}_\circ$ to be the set of graphs $\{\mathfrak{g}_a, \mathfrak{g}_b\}$ related to the two triangular facets, which is identified by the second line in Figure 8. Then, each $\mathfrak{G}_c$ contains two of the three graphs $\{\mathcal{G}, \mathfrak{g}_1, \mathfrak{g}_2\}$: the hyperplanes related to them intersect each other pairwise — they correspond to the first line in Figure 8, which implies that the simultaneous intersection of the three of them lies outside $\mathcal{P}_{\mathcal{G}}$. Hence, the canonical function can be written as

$$
\begin{aligned}
\Omega &= \left[ \frac{1}{q_{\mathcal{G}} q_{\mathfrak{g}_1}} + \frac{1}{q_{\mathfrak{g}_1} q_{\mathfrak{g}_2}} + \frac{1}{q_{\mathfrak{g}_2} q_{\mathcal{G}}} \right] \frac{1}{q_{\mathfrak{g}_a} q_{\mathfrak{g}_b}} \\
&= \left[ \frac{1}{(x_1 + x_2)(x_1 + y_a + y_b)} + \frac{1}{(x_1 + y_a + y_b)(x_2 + y_a + y_b)} \right. \\
&\quad \left. + \frac{1}{(x_2 + y_a + y_b)(x_1 + x_2)} \right] \times \frac{1}{(x_1 + x_2 + 2y_a)(x_1 + x_2 + 2y_b)}.
\end{aligned}
\tag{47}
$$

As already mentioned earlier, the fact that the intersections in Figure 8 exhaust all the zeroes of the canonical functions implies that the representations (46) and (47) are the only two representations with just physical singularities for the 1-loop 2-site contribution to the wavefunction.

**The 1-loop triangle graph**

Let us move on to a more interesting case, constituted by taking $\mathcal{G}$ to be the the 1-loop 3-site graph. The set of subgraphs identifying the intersections of the facets of the associated cosmological polytope outside of it are listed in Figure 5. The associated cosmological polytope $\mathcal{P}_{\mathcal{G}}$ lives in $\mathbb{P}^5$. Let us label with $a$, $b$, $c$ the three edges of the graph, and let $\mathfrak{g}_a$, $\mathfrak{g}_b$, $\mathfrak{g}_c$ be the subgraphs containing all the sites of $\mathcal{G}$ and all its edges but the one labelled by $a$, $b$, $c$ respectively. Let us also define $\mathfrak{g}_{ij}$ to be the subgraph containing the sites $i$ and $j$, as well as $\mathfrak{g}_i$

---

[7]In order to simplify the notation, from now on we suppress the explicit dependence of the linear polynomial $q_{\mathfrak{g}}$'s on $\mathcal{Y}$.

to be the subgraph containing the site $i$ only. Finally, it is useful to explicitly list here the set of linear polynomials $q_{\mathfrak{g}} := \mathcal{W}_I^{(\mathfrak{g})} \mathcal{Y}^I$ that determines the poles of the canonical form, choosing the local coordinates $\mathcal{Y} = (x_1, y_a, x_2, y_b, x_3, y_c)$ for $\mathbb{P}^5$:

$$
\begin{aligned}
q_{\mathcal{G}} &= \sum_{j=1}^{3} x_j, \\
q_{\mathfrak{g}_e} &= \sum_{j=1}^{3} x_j + 2 y_e, \\
q_{\mathfrak{g}_{ij}} &= \sum_{\substack{k=i,j \\ l \neq i,j}} \left( x_k + y_{e_{kl}} \right).
\end{aligned}
\tag{48}
$$

Then, we can have the following triangulations through a single subspace of the locus $\mathcal{C}(\mathcal{P}_{\mathcal{G}})$:

- $\mathfrak{G}_\circ := \{\mathcal{G}, \mathfrak{g}_1, \mathfrak{g}_2, \mathfrak{g}_3\}$ — It corresponds to the first line in Figure 5. The canonical function acquires the form

$$
\Omega = \frac{1}{q_{\mathcal{G}} q_{\mathfrak{g}_1} q_{\mathfrak{g}_2} q_{\mathfrak{g}_3}} \sum_{\{\mathfrak{G}_c\}} \prod_{\mathfrak{g} \in \mathfrak{G}_c} \frac{1}{q_{\mathfrak{g}}},
\tag{49}
$$

where $\mathfrak{G}_c$ is a set of graphs $\mathfrak{g}, \mathfrak{g}' \notin \mathfrak{G}_\circ$ which identify a codimension-2 boundary $\mathcal{P}_{\mathcal{G}} \cap \mathcal{W}^{(\mathfrak{g})} \cap \mathcal{W}^{(\mathfrak{g}')} \neq \varnothing$, and the sum in (49) runs over all such sets $\mathfrak{G}_c$. The analysis of the vertex structure of the codimension-2 faces $\mathcal{P}_{\mathcal{G}} \cap \mathcal{W}^{(\mathfrak{g})} \cap \mathcal{W}^{(\mathfrak{g}')}$ in the text, implies that the subgraphs $\mathfrak{g}, \mathfrak{g}' \in \mathfrak{G}_c$ cannot be partially overlapping: they can be either disjoint or one contained in the other, namely $\mathfrak{g}' \subset \mathfrak{g}$, in such way the number of edges departing from $\mathfrak{g}'$ is less than or equal to the number of loops of $\mathfrak{g}$. Notice that $\mathfrak{g}_a, \mathfrak{g}_b, \mathfrak{g}_c$ are all partially overlapping with respect to each other (see the first subgraphs in the second, third and fourth line of Figure 5), and the same for $\mathfrak{g}_{12}, \mathfrak{g}_{23}, \mathfrak{g}_{31}$ (see the last line in Figure 5). Hence,

$$
\{\mathfrak{G}_c\} = \{\{\mathfrak{g}_a, \mathfrak{g}_{23}\}, \{\mathfrak{g}_a, \mathfrak{g}_{31}\}, \{\mathfrak{g}_b, \mathfrak{g}_{31}\}, \{\mathfrak{g}_b, \mathfrak{g}_{12}\}, \{\mathfrak{g}_c, \mathfrak{g}_{12}\}, \{\mathfrak{g}_c, \mathfrak{g}_{23}\}\},
\tag{50}
$$

and the canonical function can be explicitly written as

$$
\Omega = \frac{1}{q_{\mathcal{G}} q_{\mathfrak{g}_1} q_{\mathfrak{g}_2} q_{\mathfrak{g}_3}} \left[ \frac{1}{q_{\mathfrak{g}_a}} \left( \frac{1}{q_{\mathfrak{g}_{23}}} + \frac{1}{q_{\mathfrak{g}_{31}}} \right) + \frac{1}{q_{\mathfrak{g}_b}} \left( \frac{1}{q_{\mathfrak{g}_{31}}} + \frac{1}{q_{\mathfrak{g}_{12}}} \right) + \frac{1}{q_{\mathfrak{g}_c}} \left( \frac{1}{q_{\mathfrak{g}_{12}}} + \frac{1}{q_{\mathfrak{g}_{23}}} \right) \right].
\tag{51}
$$

This is the OFPT representation for the 1-loop 3-site graph.

- $\mathfrak{G}_\circ := \{\mathfrak{g}_a, \mathfrak{g}_{12}, \mathfrak{g}_3\}$ – It corresponds to the second line in Figure 5. Then, the canonical function can be written as

$$
\Omega = \frac{1}{q_{\mathfrak{g}_a} q_{\mathfrak{g}_{12}} q_{\mathfrak{g}_3}} \sum_{\{\mathfrak{G}_c\}} \prod_{\mathfrak{g} \in \mathfrak{G}_c} \frac{1}{q_{\mathfrak{g}}},
\tag{52}
$$

where $\mathfrak{G}_c$ identifies a codimension-3 boundary of $\mathcal{P}_{\mathcal{G}}$ and the sum is over all possible such boundaries involving facets associated to subgraphs which do not belong to $\mathfrak{G}_\circ$. The conditions on $\mathfrak{G}_c$ are given by the vertex structure of the intersections of three hyperplanes corresponding to subgraphs on the polytope, as described in the main text. Here it is important to recall that if two of these three subgraphs are partially overlapping, a codimension-3 intersection on the $\mathcal{P}_{\mathcal{G}}$ can still exist if the third subgraph coincides

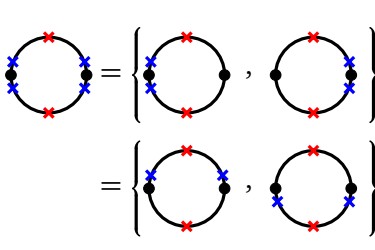

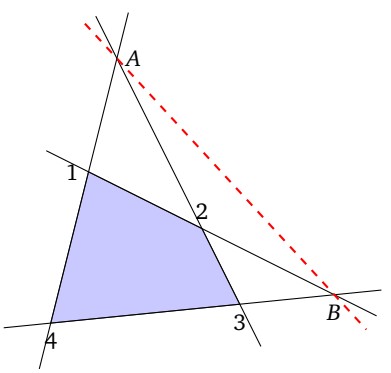

Figure 9: Intersections of the facets of the scattering facet $\mathcal{S}_\mathcal{G}$, associated to the 1-loop 2-site graph, outside $\mathcal{S}_\mathcal{G}$. On the left, their graph realisation shows that such intersections are two points. On the right, we explicitly show this scattering facet and its external triangulations. The intersections of the 1-planes containing its facets outside identify the locus of the zeroes of the canonical form (given by the red dashed line *AB*). Then, the scattering facet has two triangulations, through *A* and *B*, respectively, which do not add any additional 1-plane: $(1234) = (34A)+(A12) = (41B)+(B23)$. There are thus two representations characterised by just physical poles, and they correspond to the OFPT and the CR representations.

with the intersection of the first two subgraphs, or the intersection of one of them with the complementary of the other. Hence:

$$
\begin{aligned}
\{\mathfrak{G}_c\} = \{ & \{\mathcal{G}, \mathfrak{g}_b, \mathfrak{g}_{31}\}, \{\mathcal{G}, \mathfrak{g}_b, \mathfrak{g}_2\}, \\
& \{\mathcal{G}, \mathfrak{g}_c, \mathfrak{g}_{23}\}, \{\mathcal{G}, \mathfrak{g}_c, \mathfrak{g}_1\}, \\
& \{\mathcal{G}, \mathfrak{g}_{23}, \mathfrak{g}_2\}, \{\mathcal{G}, \mathfrak{g}_{31}, \mathfrak{g}_1\}, \\
& \{\mathfrak{g}_b, \mathfrak{g}_{31}, \mathfrak{g}_2\}, \{\mathfrak{g}_c, \mathfrak{g}_{23}, \mathfrak{g}_1\}\},
\end{aligned}
\tag{53}
$$

and the canonical function can be written explicitly as

$$
\begin{aligned}
\Omega = \frac{1}{q_{\mathfrak{g}_a} q_{\mathfrak{g}_{12}} q_{\mathfrak{g}_3}} \Bigg\{ & \frac{1}{q_\mathcal{G}} \Bigg[ \frac{1}{q_{\mathfrak{g}_b}} \left( \frac{1}{q_{\mathfrak{g}_{31}}} + \frac{1}{q_{\mathfrak{g}_2}} \right) + \frac{1}{q_{\mathfrak{g}_c}} \left( \frac{1}{q_{\mathfrak{g}_{23}}} + \frac{1}{q_{\mathfrak{g}_1}} \right) + \frac{1}{q_{\mathfrak{g}_{23}} q_{\mathfrak{g}_2}} + \frac{1}{q_{\mathfrak{g}_{31}} q_{\mathfrak{g}_1}} \Bigg] \\
& + \frac{1}{q_{\mathfrak{g}_b} q_{\mathfrak{g}_{31}} q_{\mathfrak{g}_2}} + \frac{1}{q_{\mathfrak{g}_c} q_{\mathfrak{g}_{23}} q_{\mathfrak{g}_1}} \Bigg\}.
\end{aligned}
\tag{54}
$$

There are other two representations of this type for $\mathfrak{G}_\circ = \{\mathfrak{g}_b, \mathfrak{g}_{23}, \mathfrak{g}_1\}$ and $\mathfrak{G}_\circ = \{\mathfrak{g}_c, \mathfrak{g}_{31}, \mathfrak{g}_2\}$, which corresponds to the third and fourth line in Figure 5. The canonical function in such triangulations can be obtained from (54) via a simple relabelling of the subgraphs:

$$
a \longrightarrow b \longrightarrow c, \qquad i \longrightarrow i+1 \longrightarrow i+2.
$$

To our knowledge, these representations were not known for the wavefunction coefficient related to this graph.

- $\mathfrak{G}_\circ = \{\mathfrak{g}_{12}, \mathfrak{g}_{23}, \mathfrak{g}_{31}\}$ — It corresponds to the last line in Figure 5. The signed-triangulation of the canonical function through the codimension-3 hyperplane identified by $\mathfrak{G}_\circ$, is given by

$$
\Omega = \frac{1}{q_{12} q_{23} q_{31}} \sum_{\{\mathfrak{G}_c\}} \prod_{\mathfrak{g} \in \mathfrak{G}_c} \frac{1}{q_\mathfrak{g}},
\tag{55}
$$

where, as in the previous case, $\mathfrak{G}_c$ is a codimension-3 boundary of $\mathcal{P}_\mathcal{G}$ and the sum is over all the possible sets $\mathfrak{G}_c$ whose elements are not elements of $\mathfrak{G}_\circ$. Proceeding as before, then

$$
\begin{aligned}
\{\mathfrak{G}_c\} = \{ & \{\mathcal{G}, \mathfrak{g}_a, \mathfrak{g}_1\}, \{\mathcal{G}, \mathfrak{g}_a, \mathfrak{g}_2\}, \\
& \{\mathcal{G}, \mathfrak{g}_b, \mathfrak{g}_2\}, \{\mathcal{G}, \mathfrak{g}_b, \mathfrak{g}_3\}, \\
& \{\mathcal{G}, \mathfrak{g}_c, \mathfrak{g}_3\}, \{\mathcal{G}, \mathfrak{g}_c, \mathfrak{g}_1\}, \\
& \{\mathfrak{g}_a, \mathfrak{g}_1, \mathfrak{g}_2\}, \{\mathfrak{g}_b, \mathfrak{g}_2, \mathfrak{g}_3\}, \\
& \{\mathfrak{g}_c, \mathfrak{g}_3, \mathfrak{g}_1\}, \{\mathfrak{g}_1, \mathfrak{g}_2, \mathfrak{g}_3\}\},
\end{aligned}
\tag{56}
$$

and, therefore, the explicit expression for the representation (55) is given by

$$
\begin{aligned}
\Omega = \frac{1}{q_{\mathfrak{g}_{12}} q_{\mathfrak{g}_{23}} q_{\mathfrak{g}_{31}}} \Bigg\{ & \frac{1}{q_\mathcal{G}} \Bigg[ \frac{1}{q_{\mathfrak{g}_a}} \left( \frac{1}{q_{\mathfrak{g}_1}} + \frac{1}{q_{\mathfrak{g}_2}} \right) + \frac{1}{q_{\mathfrak{g}_b}} \left( \frac{1}{q_{\mathfrak{g}_2}} + \frac{1}{q_{\mathfrak{g}_3}} \right) + \frac{1}{q_{\mathfrak{g}_c}} \left( \frac{1}{q_{\mathfrak{g}_3}} + \frac{1}{q_{\mathfrak{g}_1}} \right) \Bigg] \\
& + \frac{1}{q_{\mathfrak{g}_a} q_{\mathfrak{g}_1} q_{\mathfrak{g}_2}} + \frac{1}{q_{\mathfrak{g}_b} q_{\mathfrak{g}_2} q_{\mathfrak{g}_3}} + \frac{1}{q_{\mathfrak{g}_c} q_{\mathfrak{g}_3} q_{\mathfrak{g}_1}} + \frac{1}{q_{\mathfrak{g}_1} q_{\mathfrak{g}_2} q_{\mathfrak{g}_3}} \Bigg\}.
\end{aligned}
\tag{57}
$$

This representation for the relevant wavefunction coefficient, is also not known to our knowledge.

## C Representations for scattering amplitudes: explicit examples

In this section, we provide simple examples of triangulations of the scattering facet and explicitly write the corresponding representations for the scattering amplitudes. We consider the scattering facets $\mathcal{S}_\mathcal{G}$ related to the graphs $\mathcal{G}$ for which the representations of the wavefunction coefficients associated to triangulations of the cosmological polytope $\mathcal{P}_\mathcal{G}$, were discussed in the previous section.

**The $1$-loop $2$-site graph**

This graph contribution to the scattering amplitude is given by the canonical function of the scattering facet of the truncated tetrahedron in $\mathbb{P}^3$ discussed earlier, which is a square in $\mathbb{P}^2$. The locus of the zeroes of the canonical function is a line identified by the intersection of the opposite sides of the square (see Figure 9). At the level of the graph, such points are identified by the set of subgraphs in Figure 9.

Hence, there are just two possible triangulations, one of which provides the OFPT representation and the other the CR one

$$
\begin{aligned}
\Omega & = \frac{1}{(y_a + y_b)^2 - x_1^2} \left[ \frac{1}{2y_a} + \frac{1}{2y_b} \right] \\
& = \frac{1}{(2y_a)(2y_b)} \left[ \frac{1}{y_a + y_b + x_1} + \frac{1}{y_a + y_b - x_1} \right].
\end{aligned}
\tag{58}
$$

Notice that the factor $(2y_a \, 2y_b)^{-1}$ is the measure of the Lorentz-invariant phase space, and the CR representation in the second line makes the Steinmann relations manifest.

**The $L$-loop $2$-site graph**

Let us now consider an $L$-loop $2$-site graph, which can be obtained from the previous case by adding $L-1$ edges between its two sites. Then, such a graph has $n_s = 2$ sites and $n_e = L+1$ edges. Interestingly, the canonical form of the associated scattering facet lives in $\mathbb{P}^{n_e}$ and it has

$$\vcenter{\hbox{figure}} = \left\{\vcenter{\hbox{fig}}, \vcenter{\hbox{fig}}\right\}$$

$$= \left\{\vcenter{\hbox{fig}}, \vcenter{\hbox{fig}}, \cdots, \vcenter{\hbox{fig}}, \vcenter{\hbox{fig}}\right\}$$

Figure 10: Intersections of the facets of the scattering facet $\mathcal{S}_{\mathcal{G}}$ associated to the $L$-loop 2-site graph. There are two intersections. They are a codimension-2 and a codimension-$n_e$ hyperplane, and are depicted in the first and second line, respectively.

$\tilde{v} = n_e + 2$ facets. By $GL(1)$-invariance, its numerator is linear for any $n_e$, *i.e.* for any loop:

$$\omega(\mathcal{Y}, \mathcal{S}_{\mathcal{G}}) = \Omega\langle\mathcal{Y}d^{n_e}\mathcal{Y}\rangle = \frac{\mathfrak{n}_1(\mathcal{Y})}{\mathfrak{d}_{n_e+2}(\mathcal{Y})}\langle\mathcal{Y}d^{n_e}\mathcal{Y}\rangle. \tag{59}$$

The analysis of the compatibility among the facets presented in main text, shows that the facets of $\mathcal{S}_{\mathcal{G}}$ intersect each other outside of it in two hyperplanes, one of codimension 2 and the other of codimension $n_e$, and such two intersections fix the locus of the zeroes of the canonical form (59) (see Figure 10).

When triangulating via the codimension-2 hyperplanes, which is identified by the first line in Figure 10, the canonical function then is given by the sum of $n_e$ terms:

$$\Omega = \frac{1}{(\sum_{e\in\mathcal{E}} y_e)^2 - x_1^2} \sum_{e\in\mathcal{E}} \prod_{e'\in\mathcal{E}\setminus\{e\}} \frac{1}{2y_{e'}}. \tag{60}$$

This is nothing but the OFPT representation for the $L$-loop 2-site contribution to the scattering amplitude, which is graphically obtained by summing over all the possible ways of removing one edge.

If instead we triangulate the scattering facet via the codimension-$n_e$ hyperplane, represented by the second line in Figure 10, then the canonical function is given by the sum of just two terms[8]:

$$\Omega = \prod_{e\in\mathcal{E}}\left(\frac{1}{2y_e}\right)\left[\frac{1}{\sum_{e\in\mathcal{E}} y_e - x_1} + \frac{1}{\sum_{e\in\mathcal{E}} y_e + x_1}\right], \tag{61}$$

making manifest Steinmann relations. This structure is just the manifestation that, given a graph $\mathcal{G}$ with $n_s$ sites and $n_e$ edges, any other graph $\mathcal{G}'$ obtained from $\mathcal{G}$ by adding an edge between two sites, the causal representation for the amplitude contribution from these two graphs, has the same structure, with the same number (and type) of physical poles and differ just for the dimensionality of the Lorentz-invariant phase-space measure [25] as proved by the derivation of (40) in this paper.

As a final comment, if the graph $\mathcal{G}$ has a single edge, *i.e.* it is the 2-site line graph, then (61) returns its recursion relation expression obtained in [35] from the frequency integral representation for the propagator.

**The 1-loop triangle graph**

The scattering facet $\mathcal{S}_{\mathcal{G}}$ associated to the 1-loop triangle graph has six zeroes, all of them of codimension-3, through which we can triangulate it without introducing spurious codimension-1 boundaries (see Figure 7). Let us explicitly write down the canonical function in terms of all such signed-triangulations.

---

[8]This expression corresponds to the causal representation of an $L$-loop Feynman integral, whose loop topology contains two sites and $L + 1$ edges (see Figure 10), and was reported in [20].

- $\mathfrak{G}_\circ = \{\mathfrak{g}_j\}_{j=1}^3$ — It corresponds to the first line in Figure 7. The canonical function then acquires the form:

$$\Omega = \prod_{j=1}^3 \left( \frac{1}{q_{\mathfrak{g}_j}} \right) \sum_{\{\mathfrak{G}_c\}} \prod_{\mathfrak{g} \in \mathfrak{G}_c} \frac{1}{q_\mathfrak{g}}. \tag{62}$$

As usual, the sets $\mathfrak{G}_c$'s are determined by the compatibility conditions on the facets of $S_\mathcal{G}$ identified by all the subgraphs $\mathfrak{g} \notin \mathfrak{G}_\circ$ and they are precisely given by (50) found in the analysis of the face structure for the full cosmological polytope $\mathcal{P}_\mathcal{G}$ associated to the 1-loop 3-site graph.

The canonical function can then be explicitly written as

$$\Omega = \frac{1}{q_{\mathfrak{g}_1} q_{\mathfrak{g}_2} q_{\mathfrak{g}_3}} \left[ \frac{1}{q_{\mathfrak{g}_a}} \left( \frac{1}{q_{\mathfrak{g}_{23}}} + \frac{1}{q_{\mathfrak{g}_{31}}} \right) + \frac{1}{q_{\mathfrak{g}_b}} \left( \frac{1}{q_{\mathfrak{g}_{31}}} + \frac{1}{q_{\mathfrak{g}_{12}}} \right) + \frac{1}{q_{\mathfrak{g}_c}} \left( \frac{1}{q_{\mathfrak{g}_{12}}} + \frac{1}{q_{\mathfrak{g}_{23}}} \right) \right], \tag{63}$$

which is the OFPT representation for the 1-loop 3-site graph.

- $\mathfrak{G}_\circ := \{\mathfrak{g}_a, \mathfrak{g}_{12}, \mathfrak{g}_3\}$ — It corresponds to the second line in Figure 5. Then, the canonical function can be written as

$$\Omega = \frac{1}{q_{\mathfrak{g}_a} q_{\mathfrak{g}_{12}} q_{\mathfrak{g}_3}} \sum_{\{\mathfrak{G}_c\}} \prod_{\mathfrak{g} \in \mathfrak{G}_c} \frac{1}{q_\mathfrak{g}}, \tag{64}$$

where $\mathfrak{G}_c$ identifies a codimension-2 boundary of $S_\mathcal{G}$ and the sum is over all possible such boundaries involving facets associated to $\mathfrak{g} \notin \mathfrak{G}_\circ$. Being of codimension-2, the facets identified by the sets $\mathfrak{G}_c$'s are fixed by the compatibility conditions that the two subgraphs, say $\mathfrak{g}$ and $\mathfrak{g}'$, either can be disjoint or one can be a subgraph of the other, e.g. $\mathfrak{g}' \subset \mathfrak{g}$, but such that the number $n_{\mathfrak{g}'}$ of edges departing from $\mathfrak{g}'$ is smaller than the number $L_\mathfrak{g}$ of $\mathfrak{g}$. Such conditions fix the sets $\mathfrak{G}_c$'s to be

$$\{\mathfrak{G}_c\} = \{\{\mathfrak{g}_b, \mathfrak{g}_{31}\}, \{\mathfrak{g}_b, \mathfrak{g}_2\}, \{\mathfrak{g}_c, \mathfrak{g}_{23}\}, \{\mathfrak{g}_c, \mathfrak{g}_1\}, \{\mathfrak{g}_{23}, \mathfrak{g}_2\}, \{\mathfrak{g}_{31}, \mathfrak{g}_1\}\}, \tag{65}$$

and the canonical function can be written explicitly as

$$\Omega = \frac{1}{q_{\mathfrak{g}_a} q_{\mathfrak{g}_{12}} q_{\mathfrak{g}_3}} \left[ \frac{1}{q_{\mathfrak{g}_b}} \left( \frac{1}{q_{\mathfrak{g}_{31}}} + \frac{1}{q_{\mathfrak{g}_2}} \right) + \frac{1}{q_{\mathfrak{g}_c}} \left( \frac{1}{q_{\mathfrak{g}_{23}}} + \frac{1}{q_{\mathfrak{g}_1}} \right) + \frac{1}{q_{\mathfrak{g}_{23}} q_{\mathfrak{g}_2}} + \frac{1}{q_{\mathfrak{g}_{31}} q_{\mathfrak{g}_1}} \right]. \tag{66}$$

There are other two representation of this type, corresponding to the choices $\mathfrak{G}_\circ = \{\mathfrak{g}_b, \mathfrak{g}_{23}, \mathfrak{g}_1\}$ and $\mathfrak{G}_\circ = \{\mathfrak{g}_c, \mathfrak{g}_{31}, \mathfrak{g}_2\}$, and can be obtained from (66) via a simple relabelling. To our knowledge, these representations for this graph contribution to the scattering amplitude were not know.

- $\mathfrak{G}_\circ = \{\mathfrak{g}_{12}, \mathfrak{g}_{23}, \mathfrak{g}_{31}\}$ — It corresponds to the second-to-last line in Figure 7. The signed-triangulation of the canonical function through the codimension-3 hyperplane identified by $\mathfrak{G}_\circ$, is given by,

$$\Omega = \frac{1}{q_{12} q_{23} q_{31}} \sum_{\{\mathfrak{G}_c\}} \prod_{\mathfrak{g} \in \mathfrak{G}_c} \frac{1}{q_\mathfrak{g}}, \tag{67}$$

where, again, $\mathfrak{G}_c$ is a codimension-2 boundary of $S_\mathcal{G}$ and the sum is over all the possible sets $\mathfrak{G}_c = \{\mathfrak{g} \subset \mathcal{G} \,|\, \mathfrak{g} \notin \mathfrak{G}_\circ\}$ satisfying the compatibility conditions for the codimension-2 faces. Thus:

$$\{\mathfrak{G}_c\} = \{\{\mathfrak{g}_a, \mathfrak{g}_1\}, \{\mathfrak{g}_a, \mathfrak{g}_2\}, \{\mathfrak{g}_b, \mathfrak{g}_2\}, \{\mathfrak{g}_b, \mathfrak{g}_3\}, \{\mathfrak{g}_c, \mathfrak{g}_3\}, \{\mathfrak{g}_c, \mathfrak{g}_1\}\}, \tag{68}$$

and, therefore, the explicit expression for the representation (67) is given by

$$\Omega = \frac{1}{q_{\mathfrak{g}_{12}} q_{\mathfrak{g}_{23}} q_{\mathfrak{g}_{31}}} \left[ \frac{1}{q_{\mathfrak{g}_a}} \left( \frac{1}{q_{\mathfrak{g}_1}} + \frac{1}{q_{\mathfrak{g}_2}} \right) + \frac{1}{q_{\mathfrak{g}_b}} \left( \frac{1}{q_{\mathfrak{g}_2}} + \frac{1}{q_{\mathfrak{g}_3}} \right) + \frac{1}{q_{\mathfrak{g}_c}} \left( \frac{1}{q_{\mathfrak{g}_3}} + \frac{1}{q_{\mathfrak{g}_1}} \right) \right]. \qquad (69)$$

To our knowledge, this representation for the relevant graph contribution to the flat-space scattering amplitude is also not known.

- $\mathfrak{G}_\circ = \{\mathfrak{g}_a, \mathfrak{g}_b, \mathfrak{g}_c\}$ – It corresponds to the last line in Figure 7. For such a choice, the sets $\mathfrak{G}_c$'s are given by

$$\{\mathfrak{G}_c\} = \{\{\mathfrak{g}_{12}, \mathfrak{g}_1\}, \{\mathfrak{g}_{12}, \mathfrak{g}_2\}, \{\mathfrak{g}_{23}, \mathfrak{g}_2\}, \{\mathfrak{g}_{23}, \mathfrak{g}_3\}, \{\mathfrak{g}_{31}, \mathfrak{g}_3\}, \{\mathfrak{g}_{31}, \mathfrak{g}_1\}\}, \qquad (70)$$

and the canonical function for $\mathcal{S}_\mathcal{G}$ can be explicitly written as

$$\Omega = \frac{1}{q_{\mathfrak{g}_a} q_{\mathfrak{g}_b} q_{\mathfrak{g}_c}} \left[ \frac{1}{q_{\mathfrak{g}_{12}}} \left( \frac{1}{q_{\mathfrak{g}_1}} + \frac{1}{q_{\mathfrak{g}_2}} \right) + \frac{1}{q_{\mathfrak{g}_{23}}} \left( \frac{1}{q_{\mathfrak{g}_2}} + \frac{1}{q_{\mathfrak{g}_3}} \right) + \frac{1}{q_{\mathfrak{g}_{31}}} \left( \frac{1}{q_{\mathfrak{g}_3}} + \frac{1}{q_{\mathfrak{g}_1}} \right) \right]. \qquad (71)$$

This representation is nothing but the causal representation of [21, 25].

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
