# Peer review of "Physical Representations for Scattering Amplitudes and the Wavefunction of the Universe"

_SciPost Physics, doi:SciPost Phys. 12, 192 (2022)_

## Round 1 · Referee Report · Anonymous (Referee 1) · 2022-3-1

Report

This work extends and gives new applications for the cosmological polytope program. The authors derive several interesting results on the analytic structure of the cosmological wavefunction and in addition use the flat-space limit to translate these results into new statements about scattering amplitudes in Minkowski space. In particular, they derive new Steinmann like relations for the dS wavefunction and give a proof for the all-loop causal representation of scattering amplitudes.

There are a few minor presentation issues the authors that may be worth considering:

1) Below equation 9 and above the graphical equation there is a sentence " ... which identifies those vertices $\mathcal{Z}_i$ of $ \mathcal{P}_{g}$ such that $\mathcal{Z}_i\cdot\mathcal{W}^{(g)}$, i.e. that are not on the facet".

I believe the authors are missing the condition $\mathcal{Z}_i\cdot\mathcal{W}^{(g)}>0$ in that sentence.

2) The authors introduce the small circle marking above eqn 12 but define it later in figure 4. It may be useful to define this notation earlier.

3) It would also be useful for referencing to number the graphical equations, i.e. the graphical equation above eqn 12.

Modulo these minor issues, the paper is well-written. Although the analysis can be very technical, the authors do give a succinct review of the polytope technology. Once the above presentation issues are taken into account, I am happy to recommend this paper for publication.

Requested changes

See (1)-(3) listed above.

---

## Round 1 · Referee Report · Anonymous (Referee 2) · 2022-4-8

Strengths

  1. original
  2. it can give rise to new developments

Report

In the submitted paper the authors presented a systematic way of generating different representations for both the wavefunction of the universe in cosmology and flat-space scattering amplitudes, by making use of their invariant definition in terms of cosmological polytopes. The different representations arise from the triangulations of the cosmological polytopes and its scattering facet. Their characteristic feature is to have physical poles only. The related combinatorial description allows to consider all these representations on the same footing, recovering both the old-fashioned perturbation theory and the causal representation as well as find new ones. Moreover the authors derive new Steinmann like relations for the dS wavefunction and give a proof for the all-loop causal representation of scattering amplitudes. The paper is well presented and I recommend it for publication, in its present form.

---

## Editorial Decision

published